# Complex regulation of Gephyrin splicing is a determinant of inhibitory postsynaptic diversity

Raphaël Dos Reis[1], Etienne Kornobis [2,3], Alyssa Pereira[1], Frederic Tores[4], Judit Carrasco [5,6], Candice Gautier[7], Céline Jahannault-Talignani[1], Patrick Nitschké [4], Christian Muchardt [5], Andreas Schlosser [8], Hans Michael Maric [9], Fabrice Ango [1,10✉] & Eric Allemand [5,7,10✉]

Gephyrin (*GPHN*) regulates the clustering of postsynaptic components at inhibitory synapses and is involved in pathophysiology of neuropsychiatric disorders. Here, we uncover an extensive diversity of *GPHN* transcripts that are tightly controlled by splicing during mouse and human brain development. Proteomic analysis reveals at least a hundred isoforms of GPHN incorporated at inhibitory Glycine and gamma-aminobutyric acid A receptors containing synapses. They exhibit different localization and postsynaptic clustering properties, and altering the expression level of one isoform is sufficient to affect the number, size, and density of inhibitory synapses in cerebellar Purkinje cells. Furthermore, we discovered that splicing defects reported in neuropsychiatric disorders are carried by multiple alternative *GPHN* transcripts, demonstrating the need for a thorough analysis of the *GPHN* transcriptome in patients. Overall, we show that alternative splicing of *GPHN* is an important genetic variation to consider in neurological diseases and a determinant of the diversity of postsynaptic inhibitory synapses.

---

[1] INM, Université Montpellier, CNRS, INSERM, Montpellier, France. [2] Biomics, C2RT, Institut Pasteur, Paris, France. [3] Hub Bioinformatique et Biostatistique, Département de Biologie Computationnelle – USR 3756 CNRS, Institut Pasteur, Paris, France. [4] BIP-D Plateforme de Bioinformatique Paris-Descartes, Institut Imagine, Paris, France. [5] UMR 3738, Unité de Régulation Épigénétique, Institut Pasteur, Paris, France. [6] Max Planck Institute of Immunobiology and epigenetics, Freiburg, Germany. [7] Institut Imagine, INSERM - U1163, Unité mécanismes cellulaires et moléculaires des désordres hématologiques et implications thérapeutiques, Paris, France. [8] Institute of Structural Biology, Rudolf Virchow Center for Experimental Biomedicine, University of Würzburg, Josef-Schneider-Str. 2, 97080 Würzburg, Germany. [9] University of Würzburg, Biotechnology and Biophysics, Rudolf Virchow Zentrum Gebäude D15, Josef-Schneider-Straße 2, 97080 Würzburg, Germany. [10] These authors contributed equally: Fabrice Ango, Eric Allemand. ✉email: fabrice.ango@inserm.fr; eric.allemand@inserm.fr

As many as 60 different types of inhibitory interneurons co-exist in the brain[1,2] with distinct morphologies, physiological properties, connectivity patterns, and functions. Each of these interneurons established various heterogeneous inhibitory synapses that are arguably the quintessence of synaptic diversity. As such, it is surprising that our understanding of interneuron development and function is not supported by a better characterization of the molecules settling the diversity of their synapses. Recent studies using single-cell transcriptomic identified cell-type-specific repertoires of cell-surface and synaptic proteins expressed by various cell types, but how these factors shape the synaptic heterogeneity remain largely unknown[1,3].

GPHN is the central organizer of the inhibitory postsynaptic density (iPSD). *Gphn*-deficient mice lose the postsynaptic clustering of receptors[4–7], and therefore its activity is essential for inhibitory synaptic transmission[5,8]. GPHN harbors two protein domains, an N-terminal G-domain and a C-terminal E-domain, which are connected by an unstructured linker region[9]. Structural analysis suggests that GPHN postsynaptic clustering is formed by a network of G-domain trimers connected by E-domain dimers[10]. The linker region and the G-domain bind to multiple inhibitory synaptic molecules including GABA-$_A$ receptor subunits[11–15]. Besides the function of GPHN at inhibitory synapses, the G- and E-domains catalyze the last step of molybdenum cofactor biosynthesis[16]. Genetic alteration of this activity is associated with extensive neurological defects and early lethality[17,18]. In addition, dysfunction of GPHN-mediated neurotransmission has been implicated in severe disorders, such as Alzheimer's disease, autism, schizophrenia, epilepsy, and also in hyperekplexia[19–23].

The *Gphn* gene exhibits a complex intron-exon structure that consists of 29 exons, of which 9 are subjected to alternative splicing in tissue-specific manners[24]. Although alternative exons inclusion in *Gphn* transcripts were proposed to change the properties and functions of GPHN protein, only 2 and 16 expressed sequence tags (EST) are currently annotated for mice and humans respectively. Most significantly, patients with temporal lobe epilepsy without any genetic mutation express four abnormal *GPHN* splice variants[25]. Expressions of these irregular splice variants curtailed inhibitory synapse formation and were proposed as the main culprits for inhibitory circuit dysfunction. Yet, the precise repertoire of *GPHN* transcripts remains largely unknown and precludes our understanding of the function of GPHN at inhibitory synapses.

Here, using a targeted gene approach strategy combined with long-read sequencing, we have completely reframed the *GPHN* expression landscape and unveiled its extensive regulation by alternative splicing in mouse brain and human tissues. Such a large number of transcripts contributes significantly to the broad proteome of GPHN isoforms that interact with inhibitory Glycine and GABA-$_A$ receptors. Each *GPHN* transcript is regulated during brain development, and individual isoforms have different synaptic clustering properties. We show that inhibitory synapses harbor distinct combinations of GPHN protein isoforms, either at synapses present at specific subcellular locations or containing distinct GABA-A receptors. Changing the level of a single splice variant in the pool of other *GPHN* transcripts is sufficient to affect the number, size, and density of inhibitory synapses in the cerebellum. In addition, aberrant *GPHN* splice variants identified in neuropsychiatric diseases are expressed at low levels in the healthy brain, indicating that their fine-tuning is critical for the normal function of inhibitory synapses. Altogether, our data show that the extensive splicing regulation of *GPHN* expression is a determinant for inhibitory synapse diversity under physiological and pathological conditions

## Results

**Extensive splicing regulation of *Gphn* during brain development in mice.** To explore the repertoire of *Gphn* transcripts during mouse brain development, we performed a targeted *Gphn* RNA analysis using Third Generation Sequencing technology developed by pacific bioscience (PACBIO). This approach provides high throughput long-read sequencing of full-length transcripts, and therefore the complete analysis of exon combinations incorporated in distinct splice variants. *Gphn* cDNAs were amplified from the mouse cortex and cerebellum at four postnatal stages (Fig. 1A). Among the 114458 sequences, our custom-made bioinformatic pipeline retained 42033 high-quality circular consensus sequences (CCS; Supplementary Fig. 1). They were further grouped by clusters of identical sequences, each defining a *Gphn* alternative mRNA transcript (Supplementary Fig. 2 and Fig. 1B). In total, we identified 277 unique transcripts (*Gphn*−1 to *Gphn*−277, ordered by descending abundance; Supplementary Fig. 2 and Supplementary Table 1), demonstrating that *Gphn* is subjected to extensive alternative splicing regulation of 40 exons, especially when compared to the 8 *Gphn* alternative exons that were previously annotated in VastDB[26] (Supplementary Figs. 3 and 4). Among the 9 splicing cassettes described in the literature, splice cassettes G1, C4b, and E1 remained undetectable in our dataset, suggesting that they are expressed outside the brain and the E2 cassette was only detected in two splice variants (*Gphn*−197 and *Gphn*−181) (Supplementary Fig. 4). Our data further unveiled five entirely novel *Gphn* exons, as well as multiple exons annotated to alternative transcript(s) but, so far, without experimental validation. We also detected hundreds of new *Gphn* splice junctions, as well as several unannotated alternative 3' splice sites (ss) (Supplementary Figs. 5 and 6). Each exon is part of a myriad of different transcripts, emphasizing how difficult it is to detect a unique transcript by conventional method, in particular, to amplify a specific PCR product and assess its expression level (Supplementary Fig. 3).

Using the available public database, we built a library of 2.9 billion short-read sequences obtained from the analysis of global gene expression, and found that a significant part of the exon-exon junctions (EEJs) identified in this study were not detected previously, unveiling the importance of targeted long-read sequencing to capture the complexity of *Gphn* expression (Fig. 1C and Supplementary Table 2). A similar approach was also performed using Snaptron, a search engine indexing millions of EEJs, and only 11 of 132 *Gphn* EEJs were validated by this dataset[27]. Our data completely reframes our understanding of *Gphn* expression. In contrast to the current view that counts 20 *Gphn* exons as constitutive, we found that all internal exons were alternatively spliced. To the best of our knowledge, *Gphn* is the first gene (>10 exons) for which all internal exons are subjected to alternative splicing regulation (Supplementary Figs. 2 and 3).

Next, we analyzed the expression of specific transcripts in space and time comparing SMRT sequencing in the cerebellum and cortex at the four post-birth developmental stages (Fig. 1D). The most abundant transcript *Gphn*-1 was stably expressed in both tissues (less than 15% variation between P6 to P39). In contrast, many other transcripts displayed distinct expression profiles across developmental stages, and several of them got different regulations in cortical and cerebellar cortices. The same regulation of scaffolding protein expression has been reported in excitatory synapses using single-synapse resolution data[28]. By limiting RT-PCR assays only to transcripts containing unannotated alternative 3'ss, we validated their expression, including several that were detected with only one CCS (*Gphn*−192, −203 and −244, Fig. 1E, Supplementary Figs. 5–7A). Several variants displayed a differential expression in brain versus muscle or heart, supporting a fine-tuning tissue-specific regulation of *Gphn*

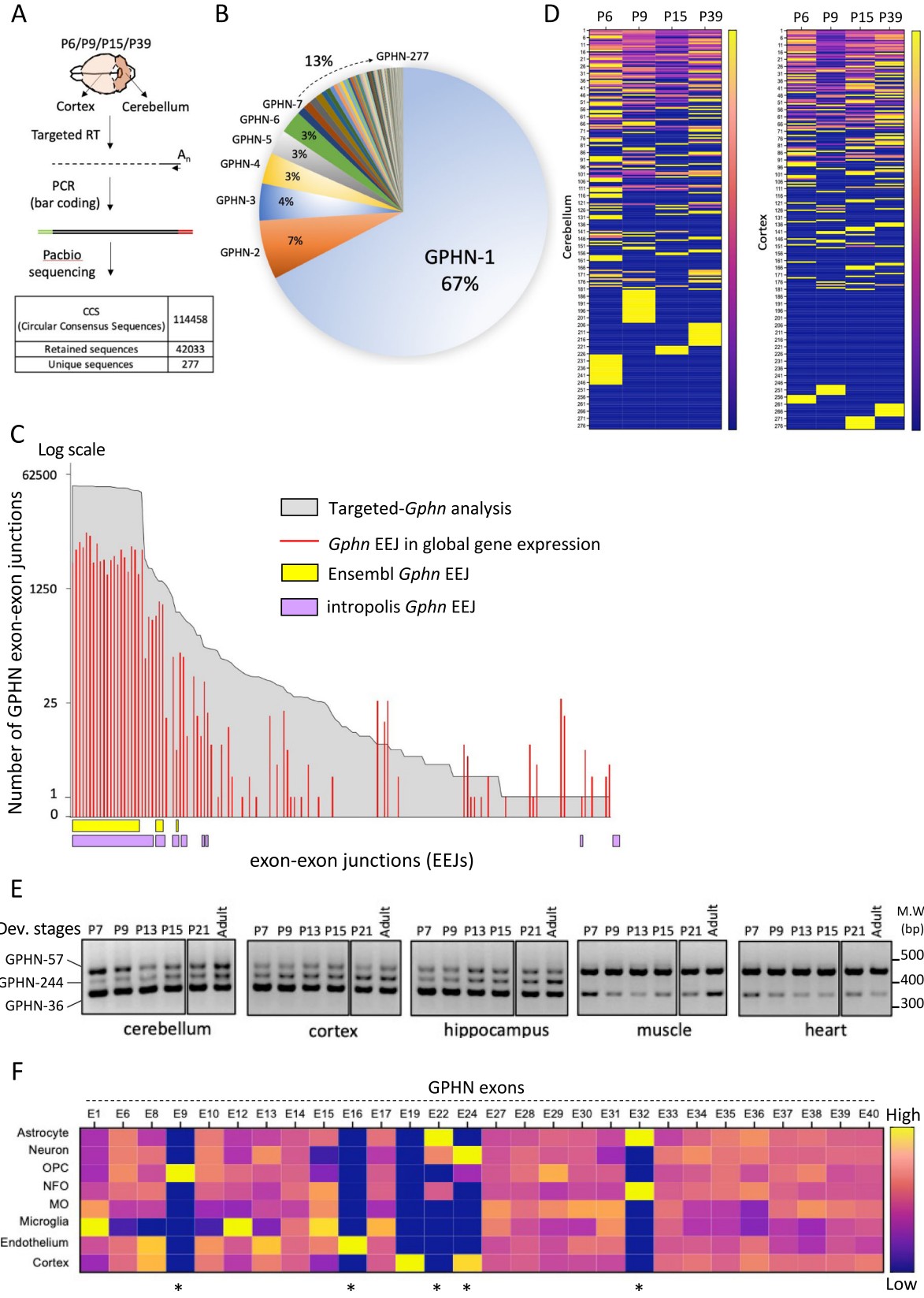

transcripts diversity. To further explore this possibility, we used publicly available RNA-seq data to expand our analysis over eight different cell types of the central nervous system, including neurons and astrocytes (Fig. 1F, Supplementary Table 3 and[29]). We detected several of the unannotated *Gphn* exons identified in our PACBIO sequencing (E9, E16, E22, E24, and E32), and acquired evidence for cell-type specific regulation of *Gphn* exons through alternative splicing in the brain. Altogether, our data redefines the *Gphn* locus with forty, rather than thirty exons as described previously (Supplementary Fig. 4) and reveals a previously unforeseen number of *Gphn* transcripts regulated during brain development.

**Fig. 1 Splicing regulation of Gphn expression in mouse brain. A** Procedure was developed to analyze *Gphn* expression by long-read sequencing. *Gphn* transcriptomes expressed at each developmental stage were prepared separately and combined in a multiplexed library sequenced with the PacBio technology. The number of retained Circular Consensus Sequences (CCS) and unique sequences are indicated at the bottom of the draw. **B** Distribution and percentages of alternative transcripts expressed by *Gphn* in Mouse brain. **C** Graph displaying the *Gphn* exon-exon junctions (EEJ) found in this study (gray), in $2,872.10^9$ short read sequences obtained by global gene expression analysis (red), in the *Ensembl* database (yellow) and using Snaptron (purple)[27]. **D** Heat map graphical representation of the 277 *Gphn* transcripts expressed at P6, P9, P15, and P39 in cortex and cerebellum tissues. Blue is associated with the lowest expression, and yellow with highest expression. **E** Example of transcripts including the selection of an alternative 3' splice site in exons 14 which are detected by RT-PCR. Amplicons corresponding to the *Gphn*−36, *Gphn*−57, and *Gphn*−244 transcripts are analyzed in 6 mouse developmental stages, 3 brain areas, skeletal muscles, and heart. **F** Quantification of *Gphn* exons expressed in various neuronal cell types of mouse brain using data from (19). Heat map graphical representation of each exon relative expression level in different cell types. Exon expression levels are displayed as the average value obtained from quadruplicate after normalization by global expression of *Gphn* in all cell types (blue for the lowest expression, and yellow for the highest expression). Stars indicated the unannotated and validated exons (Supplementary Fig. 4) and source data are provided as a Source Data file.

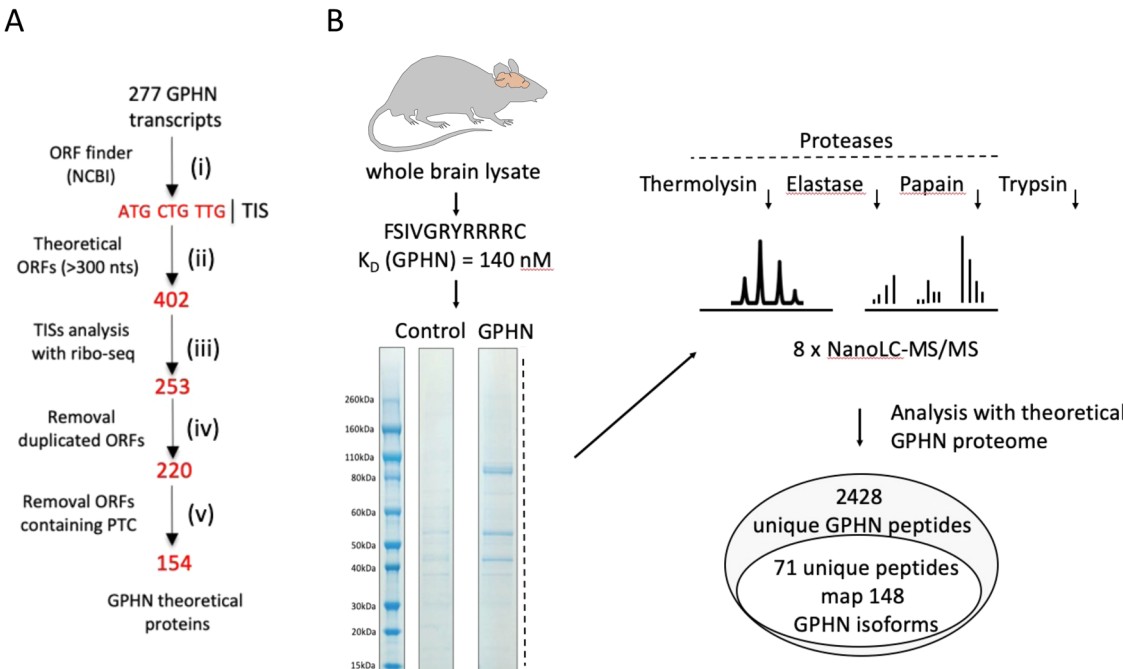

**Fig. 2 Gphn expresses a myriad of GPHN protein isoforms. A** Filtering pipeline used in this study to generate GPHN theoretical proteome, (i) Translation Initiation Start (TIS) identification, (ii) retained ORFs including at least 300 nucleotides, (iii) ORFs initiated with a TIS confirmed by biological evidences, (iv) removal of duplicated ORFs, (v) withdrawal of ORFs including PTC. **B** Workflow to isolate GPHN protein isoforms present at GABA-$_A$ and Glycine inhibitory synapses and MS processing to analyze them. GPHN was isolated from whole-brain lysate via affinity purification using a neurotransmitter receptor peptide (FSIVGRYRRRC; $K_D$ (GPHN) = 140 nM). Nano-LC MS/MS identified 2428 unique GPHN peptides, in which 71 corresponded to 148 novel GPHN protein isoforms. Source data are provided as a Source Data file.

***Gphn* splicing remodels functional domains of GPHN through a complex isoform proteome.** Whether the *Gphn* transcriptome is translated to protein isoforms or rather retained in the nucleus as "transcriptional noise" remains a critical question[30]. To tackle this question, we used purified polysomes from the mouse brain, we performed RT-PCR analysis and confirmed that novel *Gphn* transcripts are associated with the translation machinery, independently on their expression levels (Supplementary Fig. 7A). Second, we generated the theoretical proteome of GPHN in silico by tracking open reading frames (ORFs) spanning at least three hundred nucleotides (Fig. 2A and Supplementary Table 4) and filtered translation initiation starts (TISs) with ribosome profiling data as biological evidence of their activation[31]. Our approach retained 9 out of 31 potential TISs and 253 out of 402 ORFs. Interestingly, we found cases in which several alternative transcripts encoded the same ORF suggesting that multiple distinct spliced messenger RNAs could regulate a unique protein isoform. Removal of duplicated ORFs and those subjected to degradation

by nonsense-mediated mRNA decay due to existence of premature stop codon (PTC), led to list the theoretical GPHN proteome to 154 potential protein isoforms. The predicted molecular weights of the distinct GPHN isoforms are surprisingly close, suggesting that the separation of most abundant isoforms by conventional electrophoresis cannot discriminate them, thus leading to the detection of essentially a single band in western blots using anti-GPHN antibodies (Supplementary Fig. 7B, C). Third, we performed in-depth proteomic analysis of the GPHN proteins isolated from the mouse brain to interrogate existing GPHN isoforms expression from our theoretical prediction. To this end, GPHN isoforms were isolated using a functional interaction assay that pulled-down all protein isoforms capable of interacting with GABAergic and Glycinergic receptors (Fig. 2B and[32,33]). Because GPHN is known to form multimeric assembly, we expect that this approach will also isolate associated isoforms with no receptor binding site. Tandem Mass Spectrometry (MS) analysis was performed with the theoretical proteome list of

peptides resulting from the proteolytic cleavage of the GPHN ORFs and identified 2428 unique peptides that encompassed over 95% of the predicted GPHN proteome (Fig. 2B and Supplementary Table 5). In particular, the peptides corresponding to 22 undocumented exon-exon junctions were validated as well as those matching 3 new exons, and 1 uncanonical TIS (Supplementary Fig. 8). Altogether, MS provided experimental evidence supporting the expression of more than 140 GPHN isoforms in mouse brain demonstrating that the complexity of the *Gphn* transcriptome is indeed transferred to its proteome. Thus, our data reveal a myriad of GPHN isoforms expressed at the post-synaptic densities (iPSD) of inhibitory synapses.

**Distinct inhibitory synapses display a specific pattern of GPHN isoforms in cerebellum.** To study the complexity of GPHN isoforms at iPSD, a differential immunofluorescent assay was developed. Our approach used four antibodies, each recognizing different epitopes in the central region and E domain of mouse GPHN (Fig. 3A and Supplementary Fig. 9). The protein isoform GPHN-1 that harbors all the epitopes were detected by all antibodies, while other isoforms lacking one of the epitopes were differentially recognized by a specific combination of antibodies (Supplementary Fig. 10).

Using these antibodies on cerebellar slices, we found that clusters of GPHN are heterogeneously marked, indicating a diversity of incorporated epitopes which was consistent with specific combinations of GPHN isoforms (Supplementary Figs. 11 and 12). To explore whether heterogeneous GPHN clusters were synaptic. We stained synaptic sites using specific presynaptic proteins (GAD-65 or VGAT) together with GPHN antibodies in cerebellar slices of the mouse brain (Supplementary Fig. 11). Although GPHN is the ubiquitous marker of inhibitory postsynaptic sites, we found that none of the GPHN antibodies singly labeled all inhibitory synapses (Supplementary Fig. 11). We repeated this assay with combinations of anti-GPHN antibodies and found varying degrees of overlap in GPHN labeling at synapses. Our quantification further revealed the variety of GPHN epitopes combination at individual inhibitory synapses (Fig. 3B, C). Thus, *Gphn* expression is highly heterogeneous at inhibitory synapses.

One remarkable feature of inhibitory interneurons is the specific and highly structured axonal arbors with subcellular synapse specificity [e.g., dendrites, somata, or axon initial segments (AISs)] to control the input, integration, and output of their target cells. Therefore, we characterize the profiles of GPHN epitopes associated with inhibitory synapse at specific subcellular domains. For this purpose, we analyzed GPHN at Axon Initial Segments (AIS), soma and dendrites of cerebellar Purkinje cells (Fig. 3D and Supplementary Fig. 12). By combining one, two or three GPHN antibodies simultaneously, we found that subcellular domain-specific synapses harbor distinct combinations of GPHN epitopes (Fig. 3D), supporting a specific subcellular distribution of GPHN isoforms.

At the molecular level, GPHN interacts with several GABA-A and Gly receptors using a universal binding domain[33], although the molecular mechanisms that control the specific enrichment of one receptor subunit over the other remained unknown, we characterized GPHN expression pattern at selected synapses including α1, α3, or α6 GABA-AR subunits in cerebellar cortex[34–37] (Fig. 3E, Supplementary Fig. 13). We discovered a distinct pattern of GPHN epitopes at each synapse. Altogether, our data show that GPHN is differentially distributed in inhibitory synapses depending on their sub-cellular localization and the expression of subtype-specific inhibitory receptors.

**GPHN isoforms have different properties at inhibitory synapses.** The diversity of GPHN proteins unveil several isoforms for which alternative splicing deeply altered the canonic functional domains required for GPHN synaptic activities[24]. In particular, we identified isoforms with unknown functions that arbor important deletion in the G- (GPHN-5, 7 and 14) or the E-domains (GPHN-10, 28, 32, 42, and 49), as well as an insertion in the central domain (GPHN-6 and GPHN-8) (Supplementary Fig. 14A). Scarlet-tagged GPHN isoform expressions were assayed for their aggregation properties and inhibitory synapse localization in hippocampal primary neuronal culture (Supplementary Fig. 14B). Mature primary hippocampal neurons expressing isoforms were analyzed 14 days in culture after lentiviral transduction. Most isoforms that harbor the complete E domain (GPHN-1, GPHN-5, GPHN-6 and GPHN-8) formed clusters at inhibitory synapses in contrast to GPHN-14 for which expression appeared both diffused and clustered (Fig. 4A). Interestingly, skipping of exon 10 is the only difference in the exon architecture of *Gphn*−1 and −14 showing how alternative splicing can influence the expression pattern of GPHN isoforms at the synapse (Supplementary Fig. 15A). In contrast, proteins isoforms with altered E-domain (GPHN-7, GPHN-10, GPHN-28, GPHN-32, GPHN-42, and GPHN-49) were severely affected in their clustering ability and displayed a diffuse distribution in neurons (Fig. 4A). Interestingly, we noticed that these isoforms have a similar localization to GPHN fragments previously characterized for their dominant-negative activity that impair the anchoring of GABA-A receptors at the synapse[20,38].

We further analyzed the cluster properties (size, number and synaptic localization) of each GPHN isoforms able to multi-merize and found that the number and densities of these clusters at the dendrites were different (Fig. 4B–D). GPHN-1 induced the highest density of clusters, while the presence of short additional splicing cassettes in GPHN-6 and GPHN-8 reduced cluster numbers (Supplementary Fig. 15B–D). Removal of exons affecting the integrity of G domain in GPHN-5 and GPHN-14 led to a critical decrease of cluster density (Fig. 4C and Supplementary Fig. 15A, D). Interestingly, the size of clusters containing GPHN-6 and GPHN-8 were also significantly larger (~30%) suggesting that both isoforms have better multimerization properties (Fig. 4D). Because the central unstructured region has many post-translational modifications that regulate the size of GPHN clusters, it is also possible that GPHN-6 and GPHN-8 are differently regulated by post-translational modifications[39].

GPHN-5, GPHN-8 and GPHN-14 were more often localized at inhibitory synapses than GPHN-1 and GPHN-6 (Fig. 4E). We also showed that the density of inhibitory synapses in dendrites was influenced by the expression of specific GPHN isoforms in vitro (Fig. 4F), suggesting that a change in the level of a specific GPHN isoform could modulate the density of synapses.

To explore how changes in the level of GPHN isoforms affect synapse formation in vivo, we injected lentiviral constructs in the mouse brain, and quantified the number of inhibitory synapses in the cerebellum (Fig. 5A, B). Our experimental conditions led to an increase in the expression of GPHN isoform 12 days post-injection, mimicking therefore an upregulation of a specific splice variant in the pool of all alternative *Gphn* transcripts expressed endogenously. Exogenous expression of GPHN-1, GPHN-6, GPHN-10, GPHN-32 and GPHN-49 led to a similar synapse density than the control Scarlett (Fig. 5C). In contrast, GPHN-14, GPHN-28 and GPHN-42 exhibited a decreased number of synapses, while GPHN-8 enhanced the density of synapses, unveiling negative and positive dominant activities of several isoforms. These data demonstrated how the balance of *Gphn* alternative transcripts can direct the inhibitory synapse formation

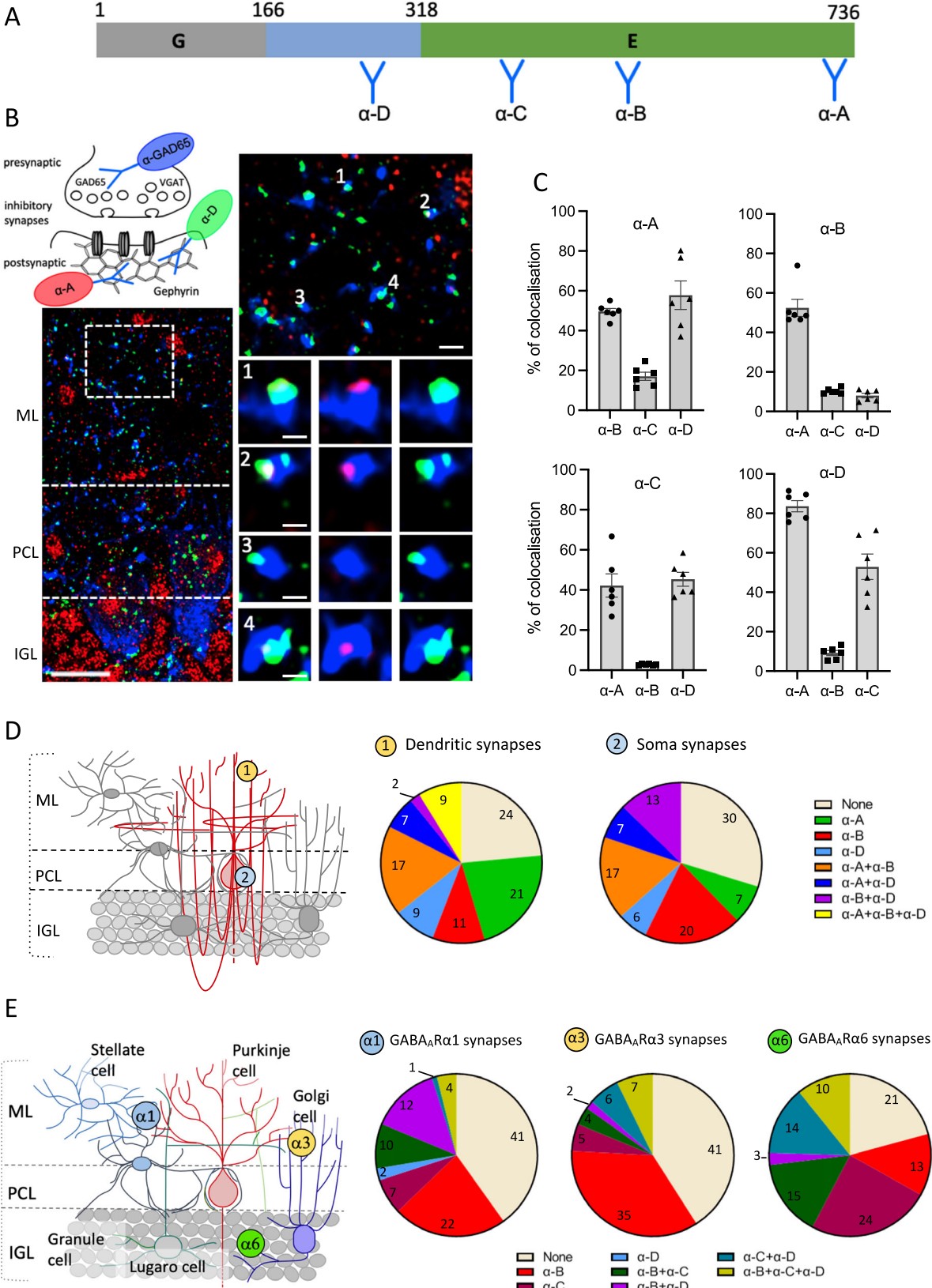

and/or maintenance in vivo. In addition, the remodeling of functional domains by alternative splicing is a mechanism to enrich selectively distinct GPHN isoforms to GABA_A circuitry and control their clustering properties. These results sustain our initial observation about the heterogeneity of endogenous GPHN epitopes and further document that distinct types of inhibitory

synapses are assembled with an array of different GPHN isoforms.

To further test that specific GPHN isoforms are heterogeneously distributed, and are not present at all synapses of an individual cell. We analyzed the distribution of GPHN-1 and GPHN-8 on the dendrite of Purkinje cells in vivo. We found that

**Fig. 3 Heterogeneous distribution of endogenous GPHN reveals diversity in the assembly of inhibitory synapses. A** Schematic of the GPHN-1 protein sequence in which are mapped the epitopes detected by four GPHN antibodies (α-A, α-B, α-C and α-D). **B** GAD-65/VGAT colocalization with GPHN epitopes α-A and α-D in cerebellar slices. Enlarged views of the white dashed box show the heterogeneous staining of synaptic GPHN epitopes. scale bars bottom-left: 15 μm, upper-right: 2 μm and 1 μm in panels 1, 2, 3, and 4. **C** Quantification of GAD-65/VGAT colocalization with multiple combinations of the GPHN antibodies was performed in $n = 6$ independent experiments. Bar graphs show means ± SEM (**D**, **E**). At the left, a schematic of cerebellar cortex cellular organization in which are displayed specific inhibitory synapses (numbered-circle) present at the molecular layer (ML), Purkinje cell layer (PCL), and internal granular layer (IGL). At the right, quantification of one or multiple GPHN epitopes at specific inhibitory synapses, for distinct neuronal sub-localization (**D**) ($n = 3$ mice), and specific subunits of GABA$_A$-R (**E**) ($n = 3$ mice). Color matches to antibodies or antibodies combinations are shown to the right (**D**) or below (**E**) the graphs. Source data is provided as a Source Data file.

the GPHN isoforms were recruited to post-synaptic sites with distinct efficiency: GPHN-1 and −8 were detected at 50% and 65% of GAD65 positive synapses respectively (Fig. 6A, B). These results confirmed our initial observations in primary cultures of hippocampal neurons, although the differences between isoforms were less pronounced (Fig. 4E). In addition, we also found that both isoforms were detected at the axon initial segment of PCs, i.e. at the site where the action potential is initiated following the integration of excitatory and inhibitory inputs (Fig. 6C). These results show that GPHN isoforms regulate inhibitory synapse properties at distinct subcellular domains. Our in vivo results corroborated our in vitro experiments (Fig. 4E) and demonstrated that specific inhibitory synapses are settled with different ratios of GPHN isoforms.

**Analysis of *GPHN* splice variant diversity is required to uncover pathological splicing regulation.** Expression of aberrant *GPHN* splice variants is associated with several neurological disorders such as epilepsy, autism, schizophrenia, Alzheimer's disease, and hyperekplexia (Table 1), prompting a deeper understanding of the regulation of *GPHN* splice variants in humans. Here, we characterized *GPHN* expression in the fetal and adult brains, cerebellum, and 18 other human tissues using targeted long-read sequencing (Fig. 7A, Supplementary Fig. 16A). We found an extensive diversity of *GPHN* transcripts in the human tissues, with 1040 unique mRNA produced from the alternative inclusion of 42 exons, some of which have tissue-specific expression profiles (Fig. 7A, Supplementary Fig. 16B, C and Table 6). Our analysis showed that 28 exons with more than 95% homology are shared between mouse and human. It is interesting to note that most of the nearly identical exons are the ones that are overrepresented in *GPHN* transcriptome (Supplementary Fig. 16B). This observation highly suggests that a significant proportion of mouse and human GPHN isoforms shared the same core protein domains. However, 12 and 14 dissimilar exons are also present in mouse and human respectively, indicating that significant differences also exist. As observed in mouse, all internal exons are alternatively included in messengers, and *GPHN* expresses a major transcript encompassing ~67% of total RNA level (Figs. 1B, 7B). Distribution analysis of splice variants highlighted 226 transcripts as a common core to all tissues, while some displayed a tissue specific expression, suggesting different cellular functions associated with distinct GPHN isoforms (Fig. 7C and Table 2). Interestingly, regardless of *GPHN* expression level, all tissues had a comparable number of *GPHN* splice variants (Fig. 7C, D), except in the brain which had the largest number of *GPHN* alternative transcripts. Among the 3 brain samples, the pool of mRNAs was very similar suggesting a dedicated regulation of *GPHN* splicing to this tissue (Supplementary Fig. 17). Moreover, the diversity of *GPHN* expression displayed a high regulation between the adult and fetal stages, but also between the cerebellum and whole brain, demonstrating an important modulation during development and across different brain areas (Fig. 7E and Supplementary Table 6). Overall, our

data revealed a much more diverse and regulated expression of human *GPHN* than the 16 alternative transcripts currently annotated in databases.

Among the *GPHN* transcriptome, we sought transcripts containing exon-exon junctions mimicking the genetic variations of *GPHN* previously characterized in patients with neurological disorders (Fig. 8A and Table 1). Surprisingly, almost all pathogenic exon-exon junctions were detected in healthy human brain samples, and we noticed that they were all included in transcripts expressed at low level, from 0.02 to 2.7% of the total *GPHN* expression (Fig. 8B). Although many of these exon-exon junctions have been classified as irregular splicing events uncovered in patients, our work demonstrates that they are actually canonic splicing events expressed in healthy individuals. Furthermore, these exon-exon junctions are present in multiple transcripts, in particular the junction between exon 5 and 12 that was detected in 82 different transcripts (Fig. 8B). We noted that the abundance and diversity of these pathogenesis-related transcripts are differently controlled through brain areas. Overall, our study shows that a detailed appreciation of the *GPHN* splice variant landscape is required to evaluate aberrant expression of GPHN isoforms in neurological diseases.

## Discussion

Despite the progress made in understanding the formation of inhibitory synapses, little is known about the molecular diversity of inhibitory synapses. Here, we demonstrate that the major scaffolding protein of inhibitory synapses, namely GPHN, is a marker of synapse diversity. *GPHN* expresses a complex landscape of transcripts tightly regulated by alternative splicing to encode a myriad of protein isoforms. We show that they are differentially assembled to specific subclasses of inhibitory synapses including those with distinct GABA-$_A$ receptors or connecting at different subcellular localization. The diversity of *GPHN* expression is highly regulated during developmental stages, between different tissues or brain areas. Among the diversity of GPHN isoforms, the change in the expression level of a single splice variant is sufficient to affect the formation/maintenance, size and density of inhibitory synapses in vivo. Furthermore, we demonstrate that aberrant *GPHN* splicing, reported in neuropsychiatric pathologies (schizophrenia, Autism Spectrum Disorders (ASDs), and epileptogenesis), are encompassed in a myriad of alternative transcripts that are expressed at low levels in brain of healthy individuals. This finding argues for further in-depth analysis of the *GPHN* transcriptome in patients to decipher what part of its expression might be affected and which isoform is clinically relevant.

**Splicing regulation of *Gphn*.** The regulation of mouse and human *GPHN* splicing must be controlled by a complex network of factors, especially because we found that all internal exons are alternatively spliced (Fig. 1F, Supplementary Figs. 2, 3, 5, and 6). Alternative splicing is catalyzed by the spliceosome, an enzyme

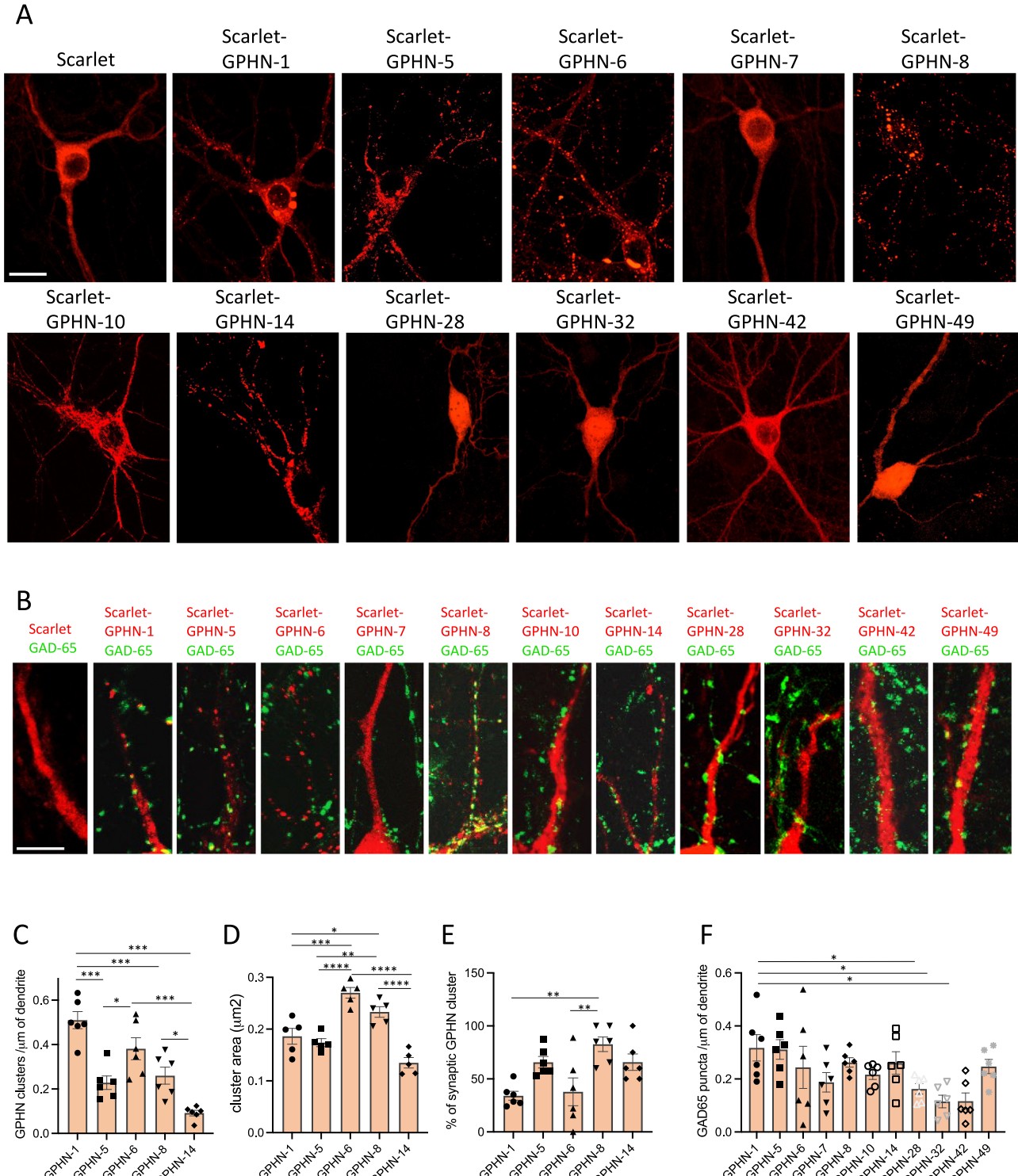

**Fig. 4 GPHN isoforms have distinct synaptic properties.** Eleven Scarlet-tagged GPHN isoforms were analyzed in hippocampal primary neuronal culture. Exogenous GPHN isoforms were selected from the pool of most expressed *Gphn* transcripts in cortex and cerebellum (Fig. 1D). **A** Representative images of global localization of GPHN isoforms (red) in neuronal cells. **B** Confocal analysis of proximal dendrites in neuronal cells expressing Scarlet or Scarlet-tagged GPHN isoforms (red), presynaptic side was stained with anti-GAD-65 (green). **C** Density of GPHN clusters along the dendrite. **D** Size of the GPHN clusters. **E** Percentage of GPHN clusters associated with a GAD-65 punctum. **F** Density of GAD-65 puncta along the dendrite. For experiments performed in C–F, at least 12 cells were analyzed in $n = 6$ independent experiments for C, E, and F, and $n = 5$ independent experiments for **D**. Bar graphs show means ± SEM. ****$P < 0{,}0001$, ***$P < 0{,}0010$, **$P < 0{,}01$, *$P < 0{,}05$. One-way analysis of variance (ANOVA) for **C–E**. *$P < 0{,}05$. Kruskal–Wallis test for (**F**). Scale bar: 25 μm (**A**) and 15 μm (**B**). Source data is provided as a Source Data file.

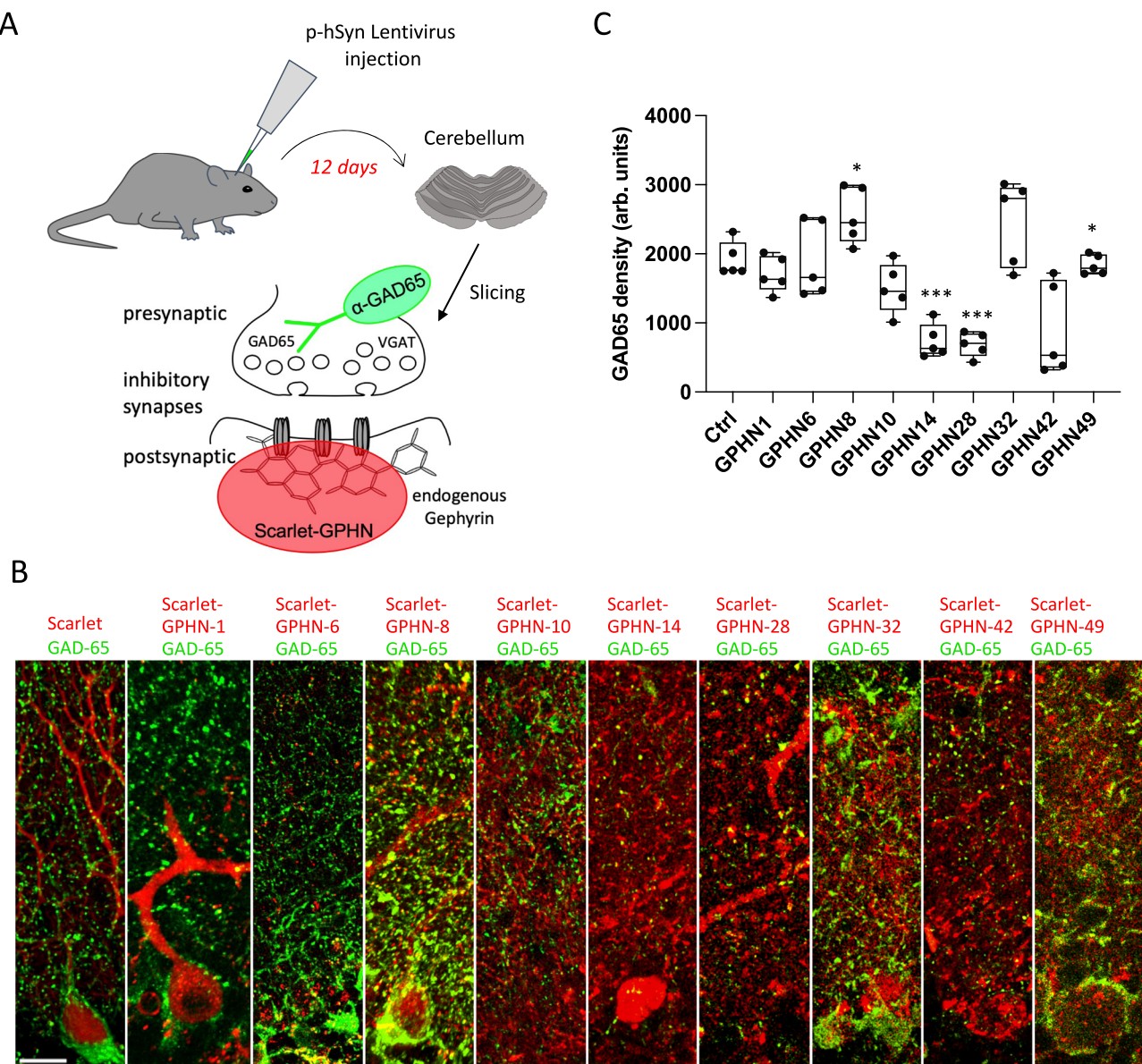

**Fig. 5 Increase of GPHN isoform levels modulates the number of inhibitory synapses in mouse cerebellum.** Analysis of inhibitory synapses connected to Purkinje cells in mouse cerebellar slices after transduction of lentivirus expressing Scarlet-tagged GPHN isoforms. **A** Schematic displaying the experimental procedure. **B** Cerebellar cortex slices in which inhibitory synapses are stained at pre- and postsynaptic sides using anti-GAD65 antibody (green) and Scarlet fluorescence (red), respectively ($n = 5$ mice for each isoform). **C** Quantification of GAD-65 density detected in panel B ($n = 5$ independent experiments). Box boundaries in **C** are the 25th and 75th percentiles, the horizontal line across the box is the median, and the whiskers indicate the minimum and maximum values. ***$P < 0,0010$, *$P = 0,0179$ and *$P = 0,0128$ for GPHN8 and GPHN49, respectively. Two-sided Student's $t$-test. Scale bar: 15 μm. Source data is provided as a Source Data file.

including more than 200 factors, as well as the influence of other gene expression machinery such as chromatin and RNA polymerase II[40]. Analysis of global gene expression in knockout mice or high-throughput screening of RNA-binding proteins has highlighted the splicing factors Nova, Sam68, PTBP2, and Rbfox as potential regulators of *GPHN* splicing[41–43]. However, it is likely that *GPHN* splicing regulators are not limited to this small set of factors. Furthermore, each human tissue is expected to have distinct regulation of *GPHN* splicing, especially in the brain where its expression is extremely diverse.

Although single-cell analysis has delineated cardinal classes of inhibitory interneurons with a specific transcriptional design to encode their synaptic communication, our study and others show an underestimation of splicing regulation[1,44–48]. Analysis of

global gene expression using high-throughput RNA-seq technologies (NGS or TGS) can only provide an overall assessment of RNAs transcribed by each gene. Indeed, the most expressed genes and the most abundant splice variants saturate by competition in the sequencing capacities, which hinders the analysis of less represented splice variants. In contrast, targeted RNA-seq analysis leads to a more qualitative analysis of gene expression reducing the intrinsic complexity of samples. Applying this strategy to study *GPHN* allowed us to unveil its complex, fine and dynamic regulation by alternative splicing. The procedure used to analyze gene expression is therefore a critical point to evaluate the portfolios of alternative mRNA isoforms. Recent studies have reached similar conclusions and demonstrated that overlooked isoforms may carry novel disease-relevant gene functions[44,49].

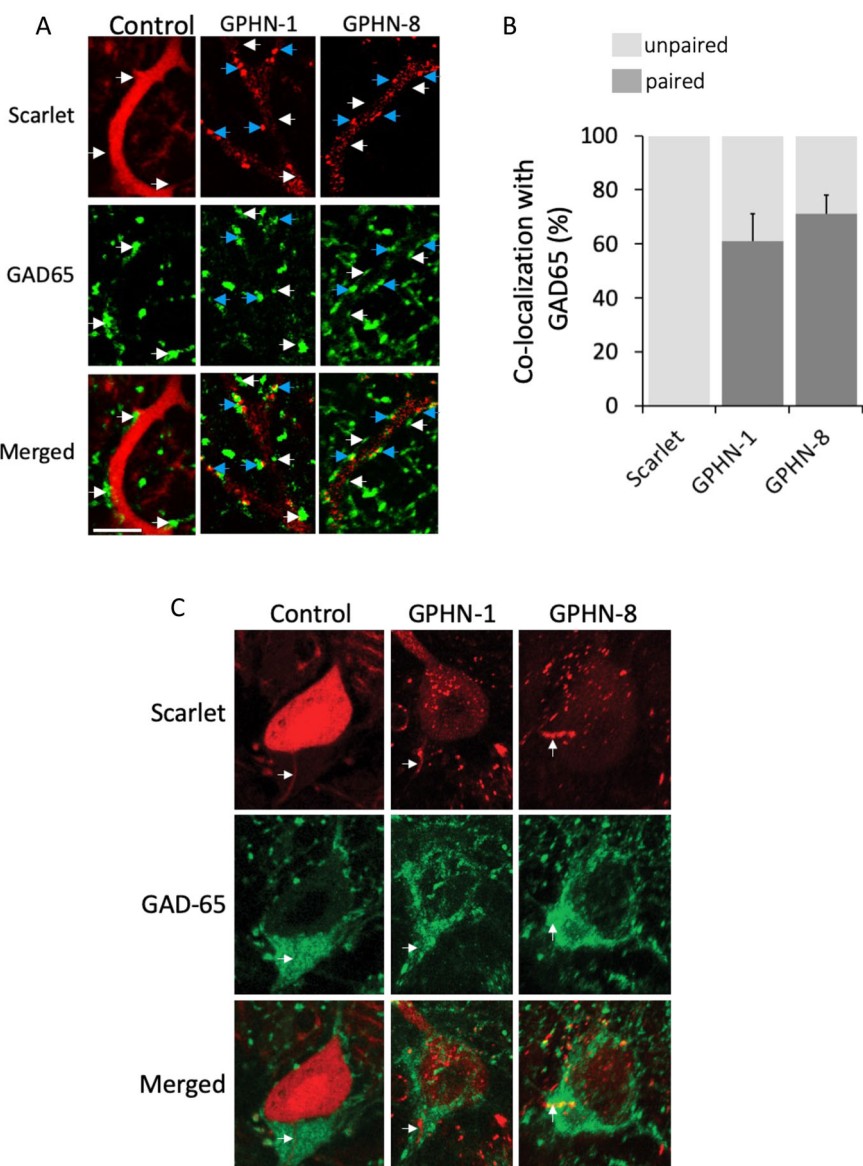

**Fig. 6 Different recruitment efficiency of GPHN isoforms to inhibitory synapses in mouse cerebellum.** Comparative analysis of two GPHN isoforms having different clustering properties in mouse cerebellar Purkinje cells. An experimental procedure was performed like in Fig. 5A using three different viruses expressing respectively exogenous Scarlet and Scarlet-tagged GPHN-1 and −8 (red). **A** Representative confocal images that display the exogenous proteins in PC dendrites of cerebellar slices. White arrows point to GAD65 synapses (green) connecting PC dendrites lacking exogenous proteins, while blue arrows point to GAD65 synapses containing postsynaptic exogenous proteins. **B** Quantification of inhibitory synapse co-localization of GAD65 and exogenous proteins ($n = 3$ independent injected animals. Bar graphs show means ± SEM. **C** Different localization of Scarlet or Scarlet-GPHN isoforms (red) at the axon initial segment (AIS) of Purkinje cells in which GAD65 is immunostained (green). Note that only GPHN-1 and −8 are detected at the AIS, while other isoforms such as GPHN-28 are not. Scale bars: 15 μm.

**Expression of GPHN isoforms shape inhibitory synapse diversity.** How inhibitory synapses diversity is regulated remains entirely unclear, in contrast to excitatory synapses, for which specificity and diversity result primarily from the combination of several postsynaptic scaffolding proteins encoded by different genes[50–55]. Here, we found that the gene encoding for the central organizer of iPSD expresses more than hundreds of isoforms, and each has a unique structural organization, and therefore distinct partner binding properties at iPSD[24]. GPHN isoforms do not localize at all inhibitory synapses and specific GABA-$_A$R containing synapses displayed different patterns of GPHN epitopes, suggesting that each synapse contains a specific ratio of GPHN isoforms. The specific GPHN isoforms distribution within a single cell could represent local activity-dependent regulation at

specific GABA-$_A$R containing synapses. Indeed, four aberrant splicing isoforms of GPHN were observed following cellular stress[25]. In addition, NOVA has been shown to regulate the alternative splicing of Gephyrin[42] and NOVA function is modulated by electrical activity[56], thus, highly suggesting that *GPHN* splicing variant could also be regulated during the change in the excitation/inhibition balance in neurons. Conversely, exons microdeletion in *GPHN* induced neuronal dysfunction leading to neuropsychiatric diseases. Several exon skipping in the G-domain is sufficient to disrupt gephyrin clusters. Disruption of gephyrin clusters was shown to decrease the amplitude and frequency of spontaneous GABAergic synaptic currents[57]. Since transient receptor-scaffold interactions govern the "diffusion trapping" of the receptors at postsynaptic sites, the distinct synaptic scaffold

**Table 1 Genetic variations and irregular splicing of *GPHN* characterized in patients with neurological disorders.**

| Diseases | Genetic variations | Previous exon nomenclature | References | Novel exon nomenclature |
|---|---|---|---|---|
| Seizure, ASD, and SCZD | Deletion | Fusion exon 3–5 | Lionel et al., 2013 | Fusion exon 10–14 |
| IGE | Deletion | Fusion exon 1–4 | Dejanovic et al. 2014 | Fusion exon 1–12 |
| IGE | Deletion | Fusion exon 4–10 | Dejanovic et al. 2014 | Fusion exon 12–27 |
| TLE | Irregular splicing | Splice junction exon 3–9 | Forstera et al., 2010 | Splice junction exon 5–16 |
| TLE | Irregular splicing | Splice junction exon 5–9 | Forstera et al., 2010 | Splice junction exon 12–16 |
| TLE | Irregular splicing | Splice junction exon 4–8 | Forstera et al., 2010 | Splice junction exon 10–15 |
| TLE | Irregular splicing | Junction exon 3–5 | Forstera et al., 2010 | Junction exon 5–12 |
| TLE | Irregular splicing | Junction exon 5–8 | Forstera et al., 2010 | Junction exon 12–15 |
| EE | Missense mutation G375D | Exon 17 G > A | Dejanovic et al., 2015 | Exon 29 G > A |

*TLE* temporal lobe epilepsy, *IGE* idiopathic generalized epilepsy, *EE* epileptic encephalopathy, *ASD* autism syndrome disorder, *SCZD* schizophrenia.
Neurological diseases are referenced in the first column and below the table, while genetic variations, previous exon annotations, references, and new exon annotations are indicated in four separated columns.

properties, distribution and dynamics of GPHN isoforms are expected to impact the synaptic strength[58,59]. With the validation of near 140 protein isoforms detected in our proteomic analysis, further analysis combining gain and loss of function strategies will be needed to explore their roles in inhibitory synapse function. This analysis is warranted by the validation of peptides spanning 22 novel exon junctions, suggesting that these GPHN isoforms have novel or altered domain organization whose functions are not yet known. The characterization of the macromolecular complexes involving GPHN isoforms with their receptors and intracellular binding partners will provide needed information to revisit iPSD molecular organization. Altogether, our study identifies GPHN isoforms as a key factor to better understand changes in inhibitory synapse function and diversity in health and diseases.

***GPHN* splicing regulation in human neuropathologies**. Characterization of pathological genetic variations in *GPHN* gene locus has so far been limited to the search for coding sequence mutations using data of exome sequencing and, more recently, whole-genome sequencing. However, ~90% of SNPs associated with traits/diseases are annotated in intronic (45%) or intergenic (43%) regions[60]. Therefore, it is likely that among these noncoding genetic variations, some dysregulate *GPHN* splicing. Either by affecting cis-splicing regulatory sequences located in introns, or by changing the expression of splicing factor that modulate *GPHN* splice variants. Targeted RNA-seq analysis of full-length transcripts should be extended in the future to analyze the expression of factors involved in the synapse plasticity, especially in the context of patients affected with neuronal disorders. This approach should unveil the genetic mechanism behind the dysregulation of highly conserved genes and open avenues for treating these diseases.

## Methods

**Mice**. G42 (GAD67-GFP) and GlyT2 (GlyT2-GFP) maintained in C57BL/6 background (C57BL/6NCrl from Charles River Laboratories) were used for immunohistochemistry experiments. Wild-type C57BL/6 were used for stereotaxic injection and immunohistochemistry experiments. Swiss mice (RjOrl:SWISS from Janvier Laboratories) were used for primary neuron cultures. Animals were kept under standard conditions with controlled temperature and lighting and received food and water ad libitum. Mice were weaned at 20–21 days of age and group-housed (less than 5 mice per cage), both male and female animals were used for all experiments. We followed the European and national regulations for the care and use of animals in order to protect vertebrate animals for experimental and other scientific purposes (Directive 86/609). All animal procedures for the characterization of Gephyrin splice variants obtained ethical approval (APAFiS #11841).

**Tissues preparation**. C57BL/6, G42, or GlyT2 mice aged 40 days were used. Animals were lightly anesthetized with isoflurane and then deeply anesthetized with sodium pentobarbital by intraperitoneal injection. Artificial CerebroSpinal Fluid (ACSF; NaCl 126 mM, KCl 3 mM, NaH$_2$PO$_4$ 1,25 mM, NaHCO$_3$ 20 mM,

MgSO$_4$ 2 mM, Dextrose 20 mM, and CaCl$_2$ 2 mM) was perfused transcardially for 2 min. The brains once extracted were placed 2 hours in 4% paraformaldehyde (PFA) in PBS. The brains were then sectioned using a microtome (Leica) to obtain 50 µm slices kept in PBS at 4 °C for 2 weeks maximum before use.

**PacBio sequencing**. RNA was extracted from mouse cortex and cerebellum tissues using a single-step method of RNA isolation by acid guanidinium thiocyanate-phenol-chloroform extraction as described[61], and a primer specific Reverse Transcription (RT) was performed with the primer GPHN-49: GTACTGTGCCTGAGGCTGC. Gphn expression was next amplified using two rounds of PCRs, the first 20 cycles with the primers GCAGTCGAACATGTAG CTGACTCAGGTCACCCACGACCATCAAATCCGTGTC and TGGATCACTT GTGCAAGCATCACATCGTAGCAGGATACAGTCAATGATATGTGGACAT GC, and after purification of PCR fragment a second PCR of 23 cycles was applied to each sample with Bar Coded (BC) primers: BC-GCAGTCGAACATGTAGCT GACTCAGGTCAC and BC-TGGATCACTTGTGCAAGCATCACATCGTAG. PCRs were purified with NucleoMag NGS Clean-up (Macherey-Nagel), quantified using Qubit, and the stoichiometric amount of each sample was finally mixed to build the multiplexed library. Library was processed as suggested by PacBio and sequenced, 114458 reads were obtained post filtering with a mean size of 16,508 bases. CCS showed a mean size of 2,3 kb.

**Oxford nanopore sequencing**. Total RNAs purified from 21 human tissues bought from Clontech (#636533 and #636643) and were subjected to reverse transcription using the specific primer GPHN-48: CAGGATACAGTCAATGA-TATGTGGACATGC. Amplification of the *GPHN* transcriptome was processed independently in each sample by two PCR steps[58]. First, a pre-amplification of 20-cycles with specific primers corresponding to the first and last exons, and then a second amplification of 18-cycles to add the barcodes. The multiplexed library was constructed using the Oxford Nanopore SQK-LSK109 kit and sequenced on MinION device.

**PacBio, ONT, and short reads analysis**
*PacBio analysis*. PacBio data were processed using informatic and manual approaches because the Isoseq 3.1.1software provided by PacBio did not provide homogeneous clusters of identical Circular Consensus Sequences (CCS). CCS were obtained from the Cold Spring Harbor sequencing platform and aligned to the Mouse genome GRCm38.p6 using "STARlong 2.5.3a [https://github.com/alexdobin/STAR]"[62] (options: -outFilterMultimapScoreRange 20 -outFilterScoreMinOverLread 0 -outFilterMatchNminOverLread 0.66 -outFilterMismatchNmax 1000 -winAnchorMultimapNmax 200 -seedSearchLmax 30 -seedSearchStartLmax 12 -seedPerReadNmax 100 -seedPerWindowNmax 100 -alignTranscriptsPerReadNmax 100000 -alignTranscriptsPerWindowNmax 10000). Genomic intervals showing alignments with the CCS (corresponding to potential exons) were extracted using "bedtools 2.27.1 [https://bedtools.readthedocs.io/en/latest/]"[63] merge (default parameters) from merged bam files generated with "samtools 1.6 [https://github.com/samtools/samtools/releases/]"[64] merge (default parameters). Each potential exon was extracted using bedtools 2.27.1 getfasta (default parameters) from Mouse GRCm38.p6 reference genome and checked by eye to keep only those framed with canonical GT-AG dinucleotides at their splice sites. Retained exons were then aligned to CCS with blastn-short algorithm of "blastn 2.2.31 [https://bioweb.pasteur.fr/packages/pack@blast+@2.2.31]"[65] (default parameters, except using an e-value threshold of 0.001 and a penalty for nucleotide mismatch of −2) to annotate each CSS for the presence or absence of distinct exon. Theoretical transcripts were constructed by concatenating the exon sequences and according to their annotation, CCS were next aligned against each theoretical transcript sequence (using a gapped global alignment, similar to the Needleman-Wunsch algorithm with "Exonerate 2.4.0 [https://github.com/nathanweeks/exonerate]", default parameters) to provide an alignment score used as quality control. Similar sequences were

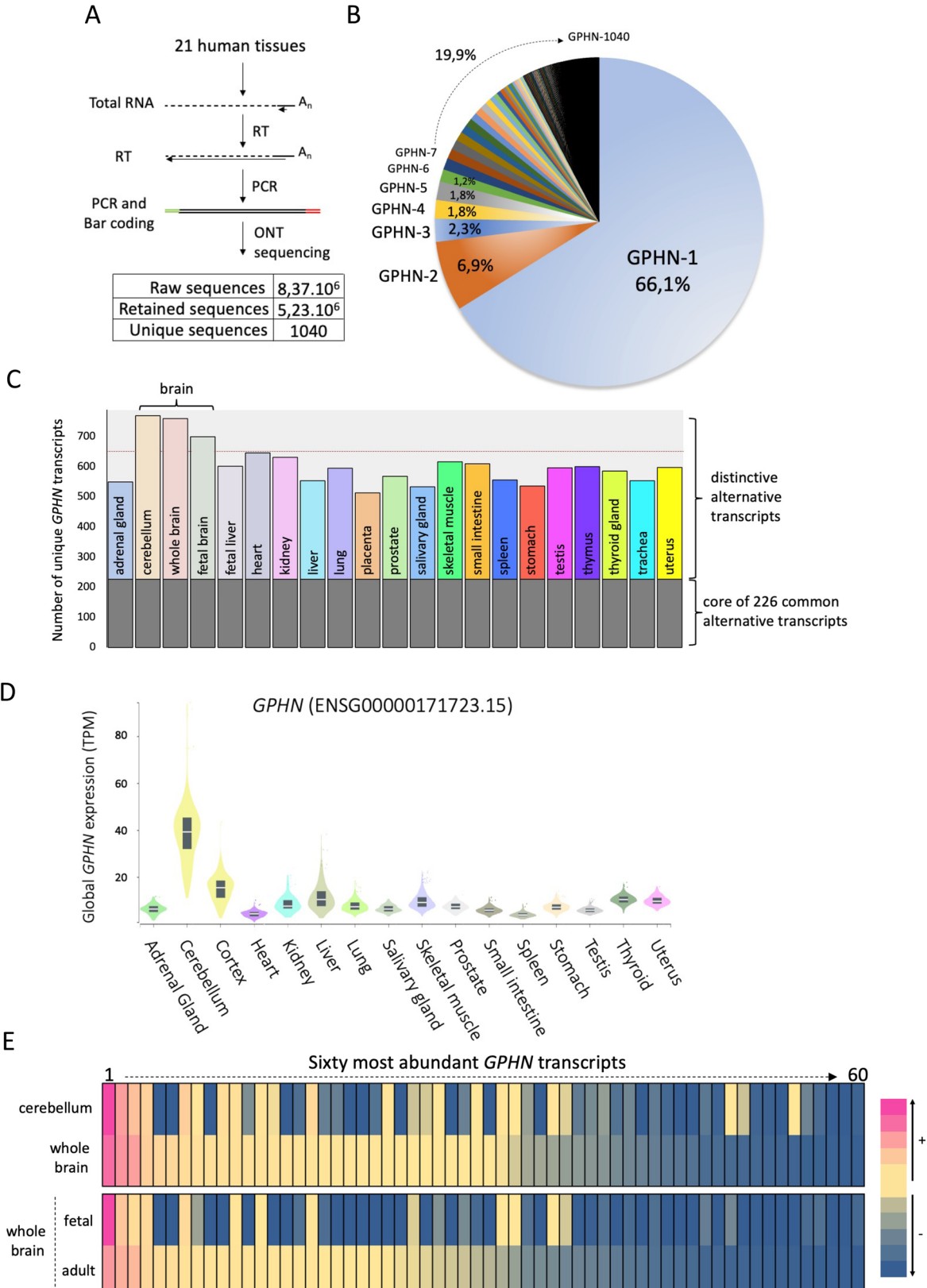

clustered together and poorly aligned CCS were removed. Homogeneity of each cluster was checked visually using 'Snapgene software [https://www.snapgene.com/]". Number of CCS present in each cluster was quantified in order to establish an overall assessment of alternative *Gphn* transcript expression. "R 3.3.2 [https://cran.rstudio.com/]" and "ggplot [https://github.com/tidyverse/ggplot2/blob/HEAD/R/plot.r]" were used to produce the overall isoform composition plot displayed Fig. 1A.

*Analysis of* Gphn *expression in brain cell types.* Gphn exons expressed in distinct neuronal cell types were analyzed using Illumina data from[29] corresponding to total gene expression. An artificial sequence containing all *Gphn* exon junctions was used to align short reads using "Hisat 2.1.0 [https://bioweb.pasteur.fr/packages/pack@hisat2@2.1.0]" (with the –no-spliced-alignment option). Using "Feature-Counts 1.6.4 [http://subread.sourceforge.net/"[65] (with options -M -fraction -largestOverlap -a exons.saf -F SAF) and the manually generated annotation of all

**Fig. 7 Splicing regulation of GPHN expression in 21 human tissues. A** Procedure and number of sequences obtained by long-read sequencing with the Oxford Nanopore technology. Amplification of *GPHN* transcriptome from each 21 human tissues was processed independently and mixed in a multiplexed library. **B** Distribution and percentages of alternative transcripts expressed by *GPHN* in the human body. **C** Graphical representation showing the proportion of splice variants that were detected in all tissues (core of common 226 alternative transcripts) and those distinctly expressed in one or more tissue(s). **D** Graphical representation of global expression level of *GPHN* in 16 distinct human tissues using the data provided by the Genotype-Tissue Expression (GTEx) project (https://gtexportal.org/home/). **E** Heat map graphical representation of the 60 most expressed *GPHN* transcripts in adult and fetal human brain, as well as the whole brain versus the cerebellum. The yellow color indicates an identical expression level between samples, while the blue scale shows a decrease and the pink an increase. Source data is provided as a Source Data file.

| Table 2 Splice variants identified in each tissue. | |
|---|---|
| | **GPHN alternative transcripts** |
| (P) Placenta | 513 |
| (FB) Fetal brain | 700 |
| (FL) Fetal liver | 602 |
| (BW) Brain (whole) | 760 |
| (C) Cerebellum | 770 |
| (SG) Salivary gland | 534 |
| (Tr) Trachea | 554 |
| (TG) Thyroid gland | 586 |
| (T) Thymus | 601 |
| (Lg) Lung | 595 |
| (H) Heart | 646 |
| (L) Liver | 553 |
| (Sp) Spleen | 556 |
| (St) Stomach | 536 |
| (AG) Adrenal gland | 550 |
| (K) Kidney | 632 |
| (SI) Small intestine | 610 |
| (Pr) Prostate | 568 |
| (U) Uterus | 598 |
| (Te) Testis | 596 |
| (SM) Skeletal muscle | 617 |

*GPHN* alternative transcripts were detected by analyzing 21 human tissues with targeted long reads sequencing. Human tissues are listed in the left column while corresponding *GPHN* splice variants are displayed in the right column.

exons in the artificial sequence, the aligned reads were quantified to assess the expression of each exon. To compare exon expression between cell subtypes, we normalized their quantification by the overall total *Gphn* expression in each cell-type.

*Analysis to validate exon-exon junction of* Gphn *expression in mouse brain. Gphn* exon junctions detected using short reads reported in 82 publicly available data sets (listed in Supplementary Table 1). An artificial sequence containing all exon junctions found in Fig. 1A was used to align short reads using Hisat 2.1.0 (with the –no-spliced-alignment option). Using FeatureCounts 1.6.4[65] (-minOverlap 15) and the manually generated annotation of all exons junctions in the artificial sequence (after calculating the minimum size of the exon-exon junction for them to be unique i.e. 9 bases 5' of junction and 6 bases 3' of junction) the aligned reads were quantified to assess junction detection. Finally, we compared them with read counts supporting each junction obtained by PacBio sequencing.

*ONT analysis.* Raw data were processed for base calling using "GUPPY [https://community.nanoporetech.com/downloads] and long-read sequences were analyzed with a homemade pipeline. Demultiplexing of samples was processed similarly to the PacBio analysis and individual sets of raw sequences were obtained for each human tissue sample (149 K up to 397 K). Using "LAST version 1205 [https://gitlab.com/mcfrith/last]"[67], the sequences were aligned on Human Genome GRCh38 and the corresponding BAM files were reviewed manually to annotate all exons. Exons were used to produce a universal reference and all reads were aligned on this particular reference with LAST. Using standard genomic tools, samtools[62], bedtools[68], a specific transcript barcode was generated to each sequence depending on its exon architecture, then pooled in clusters defining each alternative *GPHN* transcripts. Given the library preparation, we only retained the sequence containing at least the first and last exons in their barcode. Finally, we only retained transcripts seen 10 times or more in tissues.

The mouse and human GPHN exons architecture (Supplementary Fig. 5C) were compared by aligning mouse features onto the human features using LAST.

**DNA constructs**. The cDNAs corresponding to selected *Gphn* alternative transcripts were built using QuikChange II Site-Directed Mutagenesis Kit and cloned in a modified version of pLVX-puro vector for which CMV promoter was replaced by human Syn promoter. Each cDNA is cloned in 5' fusion with Scarlet and Flag-V5 tags. Lentiviral particles were produced using HEK-293 cells (ATCC; 293 T # CRL-3216) co-transfected with pLVX-FV5-Scarlet-GPHNs, p8.7, and pCMV-VSV-G, then concentrated by gradient centrifugation. All constructs used in this study are listed in Supplementary Fig. 4B and available on request. The expression of exogenous proteins were probed by western blot using anti-V5 (Invitrogen).

**In silico GPHN theoretical proteome**. The following description is summarized in Fig. 2A and reported in Supplementary Table 4. Each cDNA was processed with "ORFfinder [https://www.ncbi.nlm.nih.gov/orffinder/]" using a minimal ORF length of 300 nucleotides, "ATG" and alternative initiation codons parameters. The 31 translation initiation sites were next filtered with data published in[31] to retain only those supporting biological evidence. ORFs found more than once were removed, as well as those containing a stop codon in exons upstream of exon 40 (such stop codon was considered as a PTC). ORFs obtained from this pipeline were named as the theoretical GPHN proteome. Protein extracts from cerebellum, cortex, hippocampus, muscle, heart, and spleen were prepared and separated in PAGE-SDS to probe endogenous GPHNs by western blots using mab3B11.

**RT-PCR analysis**. RNA extracted from cortex and cerebellum tissues at P7, P9, P13, P15, P21, and P40 were retro-transcribed with GPHN-49 (see PacBio library) and amplified by PCR using specific set of primers:

Supplementary Fig. 5A: GGAGTCCTCACAGCCCACATAAAC and CTAGCCACCTTGGTGATATCTACAGC
Supplementary Fig. 5B: CTCTTGCTGCAAAGTTGACCAACTTTAG and GTTCCTTTGGCCAAAACACACTC
Supplementary Fig. 6A: CAAGAAAGGATCTCAGTAGTGCAAGTTG and CTGATACCCTCATTCAAGGCATTGAG
Supplementary Fig. 6B: GTCCTCACAGTGGTTGCCG and CTGCATCTTTCTCAGTGCAGGTACAAC
Supplementary Fig. 6C: GAGTCCTCACAGGAAAGATTCGGG and GGTGATGCCAAGTCAGTATACACC

Finally, PCRs were analyzed in agarose gel and stained with ethidium bromide.

**Polysome preparation**. Translation machinery was prepared from adult mouse brains as described[66]. Briefly, mice are anesthetized with an overdose of Xylazin and Ketamine in NaCl and transcardially perfused with 20 mL of Hank's balanced salts solution containing 200 μg/mL of cycloheximide to immobilize ribosomes on associated transcripts. Brains were rapidly extracted to prepare cytoplasmic extracts, which were then treated or not with EDTA and separated on a sucrose gradient. RNAs were finally purified after dilution of sucrose fractions with $H_2O$ (50% v/v). Ribosomal RNAs (28 S, 18 S, and 5 S) were separated in agarose gel stained with ethidium bromide, while *Gphn* isoforms were amplified and analyzed as Supplementary Fig. 5A.

**Preparation of whole-brain lysates**. After cervical dislocation, whole brains from adult male/female C57BL6 wild-type mice were removed from the skull and rapidly cooled to 4 °C and homogenized in 1 ml lysate buffer (20 mM HEPES, 100 mM KCH3COOH, 40 mM KCl, 5 mM EGTA, 5 mM MgCl2, 5 mM DTT, 1 mM PMSF, 1% Triton X, completeTM EDTA-free protease inhibitor cocktail (Roche, Mannheim, Germany), pH 7.2 per 200 mg using a pistol homogenizer (8 strokes at 900 rpm). The homogenate was centrifuged at 10,000 × g and 4 °C for 15 min. The supernatant was aliquoted and flash frozen in liquid nitrogen and stored at −80 °C.

**Gephyrin isolation from brain lysate**. The resin for gephyrin pulldown was prepared in 400 μl micro-spin columns (Thermo Scientific, Karlsruhe, Germany). The Peptide FSIVGRYPRRRRC ($K_D$ (Gephyrin) = 140 pM) (10.1038/nCHeM-BIO.2246) was dissolved in coupling buffer (50 mM Tris, 5 mM EDTA, pH 8.5) at a concentration of 1 mM and incubated for 2 h at RT with washed and equilibrated

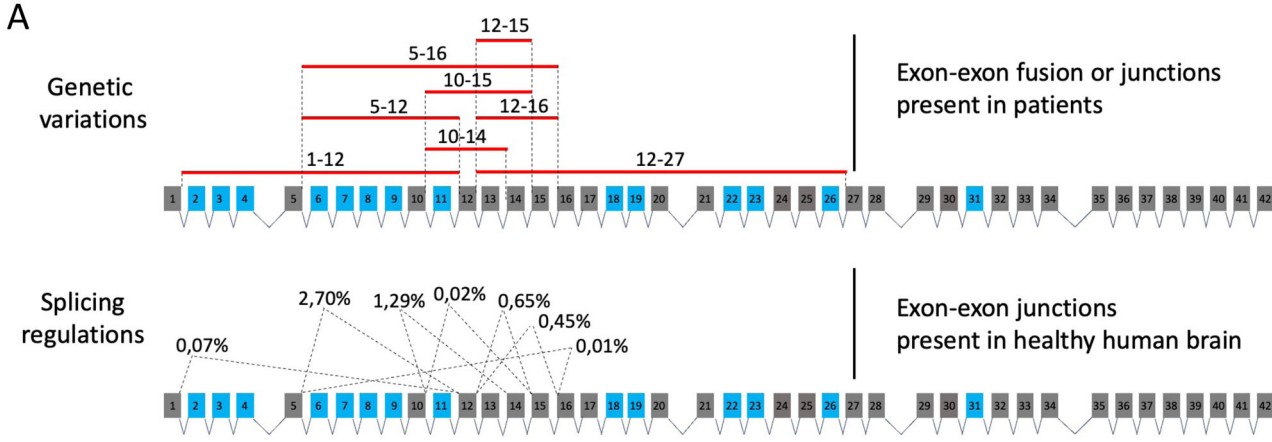

**Fig. 8 Expression of pathological GPHN transcripts is repressed in the healthy human brain. A** Schematic representation of genetic variations found in patients (upper panel) and the corresponding spliced exon-exon junctions identified by this study in the healthy human brain samples (lower panel). Percentages of each splicing event indicated the average detection in the adult entire brain, fetal entire brain, and adult cerebellum. **B** Table summarizing the detection of each exon-exon junctions in brain samples, and the number of unique transcripts containing them.

| novel exon nomenclature (this study) | whole brain | | fetal brain | | adult cerebellum | |
|---|---|---|---|---|---|---|
| | % of total *GPHN* expression | unique transcripts | % of total *GPHN* expression | unique transcripts | % of total *GPHN* expression | unique transcripts |
| fusion exon 10-14 | 3,276 | 59 | 0,313 | 17 | 0,294 | 14 |
| fusion exon 1-12 | 0,057 | 5 | 0,089 | 9 | 0,055 | 4 |
| fusion exon 12-27 | ND | ND | ND | ND | ND | ND |
| EEJ 5-16 | 0,011 | 2 | 0,013 | 1 | 0,016 | 2 |
| EEJ 12-16 | 0,281 | 12 | 0,804 | 21 | 0,271 | 11 |
| EEJ 10-15 | 0,024 | 2 | 0,021 | 1 | 0,012 | 1 |
| EEJ 5-12 | 2,927 | 82 | 2,042 | 73 | 3,127 | 73 |
| EEJ 12-15 | 0,543 | 29 | 0,752 | 33 | 0,651 | 28 |
| exon 29 G>A | ND | ND | ND | ND | ND | ND |

Ultra Link iodoacetyl resin (Thermo Scientific, Karlsruhe, Germany). The resulting resin was washed and residual iodoacetyl groups were quenched by incubation with 1 mM Cysteine for 2 h and stored at 4 °C after extensive washing with a coupling buffer.

Whole-brain lysate was incubated on the prepared FSIVGRYPRRRRC-resin for 30 min at 4 °C and washed three times with lysate buffer. Bound proteins were eluted with NuPAGE LDS sample buffer (Life Technologies) at 90 °C for 3 min. Tris(2-carboxyethyl)phosphin (TCEP) was added to a final concentration of 20 mM and incubated at 90 °C to achieve complete reduction. Subsequently, proteins were alkylated using iodoacetamide (120 mM) for 20 min at room temperature in the dark, 30 μl of the resulting denatured, reduced and alkylated proteins were separated by mass by NuPAGE Novex 4–12% Bis-Tris gels (Life Technologies) with MOPS buffer. Gels were washed three times for 5 min with water, stained for 45 min with Simply Blue™ Safe Stain (Life Technologies) and washed with water for 1 h.

Gel bands were excised and destained with 30% acetonitrile in 0.1 M $NH_4HCO_3$ (pH 8), shrunk with 100% acetonitrile, and dried in a vacuum concentrator (Concentrator 5301, Eppendorf, Germany). Digests were performed with 0.1 μg protease (trypsin, elastase, thermolysin, or papain) per gel band overnight at 37 °C in 0.1 M $NH_4HCO_3$ (pH 8). After removing the supernatant, peptides were extracted from the gel slices with 5% formic acid, and extracted peptides were pooled with the supernatant.

**NanoLC-MS/MS analysis.** NanoLC-MS/MS analyses were performed on an Orbitrap Fusion (Thermo Scientific) equipped with an EASY-Spray Ion Source and coupled to an EASY-nLC 1000 (Thermo Scientific). Peptides were loaded on a trapping column (2 cm × 75 μm ID, PepMap C18, 3 μm particles, 100 Å pore size) and separated on an EASY-Spray column (25 cm × 75 μm ID, PepMap C18, 2 μm

particles, 100 Å pore size) with a 30-min linear gradient from 3 % to 40% acetonitrile and 0.1% formic acid.

Both MS and MS/MS scans were acquired in the Orbitrap analyzer with a resolution of 60,000 for MS scans and 15,000 for MS/MS scans. HCD fragmentation with 35% normalized collision energy was applied. A top speed data-dependent MS/MS method with a fixed cycle time of 3 s was used. Dynamic exclusion was applied with a repeat count of 1 and an exclusion duration of 30 s; singly charged precursors were excluded from selection. Minimum signal threshold for precursor selection was set to 50,000. Predictive AGC was used with AGC a target value of 2e5 for MS scans and 5e4 for MS/MS scans. EASY-IC was used for internal calibration.

**Data analysis.** MS data were analyzed with PEAKS Studio X+ (Bioinformatics Solutions Inc., Canada). Raw data refinement was performed with the following settings: Merge Options: no merge, Precursor Options: corrected, Charge Options: 1–6, Filter Options: no filter, Process: true, Default: true, Associate Chimera: yes. De novo sequencing and database searching were performed with a Parent Mass Error Tolerance of 10 ppm. Fragment Mass Error Tolerance was set to 0.02 Da, and Enzyme was set to none. The following variable modifications have been used: Oxidation (M), pyro-Glu from Q (N-term Q), phosphorylation (STY), acetylation (protein N-terminal), and carbamidomethylation (C). A maximum of three variable PTMs were allowed per peptide.

Data were searched against a fasta database concatenated from UniProt_mouse (UP000000589, reference proteome, 55408 proteins, 5-Nov-2019, all variants), predicted (XP_) and validated (NP_) gephyrin sequences from NCBI and a custom ORF database constructed as described above. The list of identified peptides was filtered to 1% PSM-FDR. All gephyrin-derived peptides were matched to the entries of the ORF database using ProteoMapper[69].

## Immunohistochemistry

*Free-floating immunochemistry*. Slices were incubated in a blocking solution (0.1% Triton, 5% Horse Serum in Tris Buffer Saline (TBS) for 2 hours. The solution was then replaced by the primary antibody solution (0.1% Triton, 5% Horse Serum in TBS) with antibodies added at the indicated dilution, for an overnight incubation. The slices were washed three times for 10 min with phosphate buffer saline (PBS), before the incubation in a solution (0.02% Triton, 5% Horse Serum in TBS) containing the secondary antibodies for 1 hour. The slices were again washed three times for 10 min with PBS. The slices were then mounted between slide and coverslip using a hardening mounting medium (Vectashield Hard Set). Please refer to our "antibody table" in Supplementary Information for primary and secondary antibodies used in this study.

## Hippocampus primary cell culture

Swiss mice at 18 days of gestation were deeply anesthetized with isoflurane and then euthanized by cervical dislocation. The pups were decapitated and the heads were placed in ice-cold PBS supplemented with glucose and antibiotics (Penicillin-Streptomycin). The brains were extracted and the hippocampi dissected and placed in 37 °C Neurobasal medium (Gibco). The hippocampi were then dissociated with three glass-pasteur pipettes of decreasing tip widths. The suspension was then centrifuged at $150 \times g$ for 8 min, the supernatant discarded and the pellet re-suspended in 37 °C Neurobasal supplemented with B27, L-glutamine and antibiotics. The neurons were plated onto poly-L-ornithine coated coverslips and cultured for 11 days in an incubator at 37 °C and 5% $CO_2$.

## Cell culture infection

One day after the plating, the neuron cultures were infected using 1 μL of a suspension of lentivirus containing the relevant plasmid. The neurons were cultured for 10 more days in the conditions described previously.

## Cerebellar stereotaxic injection

40–50 day old C57BL/6 male mice were used. Mice were deeply anesthetized with isoflurane before being placed on a stereotaxic frame fitted with a custom-made isoflurane mask. Ophthalmic gel (Ocry-gel) was placed onto the mice's eyes and subcutaneous injection of lidocaine (Xylocaine) in the area was made prior to the incision of the skin. Craniotomy was performed with a fine drill bilaterally. Glass capillaries were filled with lentivirus suspensions and lowered into the cerebellar cortex. Injections were performed using a Nano-Liter Injector and UMP3000 micro pump and controller (World Precision Instruments), 1 μL per hole was dispensed over a three minutes injection. The capillary was left in place for three minutes and then slowly pulled out over a two minutes period. Tissues were then glued using surgical skin glue (Vetbond). Mice were given buprenorphine (Vetergesic) for analgesia. Animals were kept 10 days before brains were collected for immunohistochemistry assay.

## Image acquisition and quantification

Tissue and culture samples were imaged on a Zeiss LSM780 confocal microscope to obtain a stack; pinhole, detection filter settings, dwell time and step size were kept identical (x40 oil immersion objective, 1.3 NA, 1024 × 1024 resolution, 16 bits, 0.4 μm step size). Photomicrographs were obtained with the following band-pass and long-pass filter setting: alexa fluor 405 (Band pass filter: 460/50), alexa fluor 488/Cy2 (band pass filter: 505–530), Cy3 (band pass filter: 560–615) and Cy5 (long-pass filter 650). All parameters were held constant for all sections from the same experiment. At least three slices per injection site were used in all immunofluorescence analyses ($n = 3$ mice/staining).

Hippocampus neuron cultures were repeated at least three times, and infected neurons were analyzed. For each neuron, proximal dendrites (50–100 μm) were used as ROI. Analyses were made with Fiji (ImageJ) software. Images were thresholded using MaxEntropy Auto-Threshold command. GPHN clusters or GAD-65 puncta were annotated and counted. Measure Particle function was used to measure GPHN cluster area.

For immunofluorescence analyses at least three mice were used and at least three slices per mice were imaged. For each image, ROI of identical size was selected within the region of interest. For cerebellar slices, images were taken so as to contain all three layers. All analyses were made either at the molecular layer, the Purkinje cell layer or granule cell layers to evaluate layer-specific GPHN localization. For HEK cell cultures, staining was repeated three times, one image containing several cells were taken for each experiment.

Colocalization between clusters was made with Icy software using the SODA plugin (Statistical Object Distance Analysis) when the number of studied channels was below three. ROI of the relevant areas were delineated onto the images, and the SODA 2 colors or SODA 3 colors protocol was applied. Results were checked once by manual counting using the method explained below to ensure threshold and distance parameters were adequate by two experimenters.

For images with four channels and for the verification of SODA analyses, Fiji (ImageJ) software was used. Images were thresholded using MaxEntropy Auto-Threshold command. Within a ROI, all clusters from each channel were counted and annotated, and clusters were deemed colocalized when the distance separating them was below 210 nm which is the optical resolution of the confocal with the x40 objective. The analysis was done independently by two experimenters.

## Statistical analysis

All statistical analyses were performed using GraphPad Prism 8 (GraphPad Software). For quantitative analysis, no sample size calculation was performed. Experiments were designed with at least three independent biological replicates, as an accepted standard procedure in the field. Normality was assessed with the Shapiro–Wilk normality test. Homoscedasticity was assessed with the Barlett's test. Parametric data with the same standard deviation were analyzed by *t*-test, one-way ANOVA, or two-way ANOVA followed by the comparison of multiple samples with Tukey post hoc analysis. Parametric data with significantly different standard deviation were analyzed by Welch's ANOVA followed by comparison of multiple samples with Dunett's T3 post hoc analysis. Non-parametric data were analyzed by the Kruskal–Wallis one-way analysis of variance on ranks followed by the comparison of multiple samples with Dunn post hoc analysis. $P$ values <0.05 were considered statistically significant. $*p < 0.05$, $**p < 0.01$, $***p < 0.001$, $****p < 0.0001$. ns, not significant. Data are presented as mean ± SEM.

**Reporting summary**. Further information on research design is available in the Nature Research Reporting Summary linked to this article.

## Data availability

Fastq files of PacBio and ONT sequencing data generated in this study have been deposited in the ENA database (https://www.ebi.ac.uk/ena/browser) under accession code PRJEB49417 and PRJEB49420, respectively. The mass spectrometry proteomics data have been deposited to the ProteomeXchange Consortium via the PRIDE partner repository (https://www.ebi.ac.uk/pride/) with the dataset identifier PXD033751. BED files corresponding to GPHN exons annotated in mm39 and Hg38 are provided in supplemental data. BAM files corresponding to all GPHN alternative transcripts identified in mouse and human samples are provided in supplemental data. Several dataset available in public deposit were used for re-analyzing, we list below their accession numbers: *Gphn* expression in specific brain cells (Fig. 1F):. https://www.ncbi.nlm.nih.gov/geo/query/acc.cgi: GSM1269903, GSM1269904, GSM1269905, GSM1269906, GSM1269907, GSM1269908, GSM1269909, GSM1269910, GSM1269911, GSM1269912, GSM1269913, GSM1269914, GSM1269915, GSM1269916, GSM1269917, GSM1269918, and GSM1269919. Detection of *Gphn* exon-exon junctions using previous dataset (Fig. 1C):. https://www.ncbi.nlm.nih.gov/sra/ SRX026442, SRX026441, SRX026440, SRX026439, SRX026426, SRX026425, SRX026424, SRX026423, SRX2498779, SRX2498778, SRX2487430, SRX2487429, SRX2370417, SRX2370416, SRX2236736, SRX2236735, SRX2236734, SRX1923016, SRX1923015, SRX1923014, SRX1923013, SRX1923012, SRX1923011, SRX1923010, SRX1923009, SRX1923008, SRX1923007, SRX1923006, SRX1923005, SRX1923004, SRX1923003, SRX1923002, SRX1923001, SRX1838141, SRX1838140, SRX1838139, SRX1838138, SRX1829019, SRX1829018, SRX1829017, SRX1715815, SRX1715814, SRX1715813, SRX1715809, SRX1715808, SRX1715807, SRX1715803, SRX1715802, SRX1715801, SRX1715797, SRX1715796, SRX1715795, SRX1701411, SRX1701410, SRX1701408, SRX1701407, SRX1680961, SRX1680960, SRX1680959, SRX1620344, SRX1620344, SRX1620344, SRX1620344, SRX1620344, SRX1620344, SRX1620343, SRX1620343, SRX1620343, SRX1620343, SRX1620343, SRX1620343, SRX1620342, SRX1620342, SRX1620342, SRX1620342, SRX1620342, SRX1620342, SRX1607437, SRX1607436, SRX1607435, SRX1607434, and SRX1607433. Source data are provided with this paper.

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

## Acknowledgements

We thank Y. Lallemand and Y. Porozhan for their help in perfusing mouse for ribosomal fractionation; the iExplore and RIO-Imaging Platforms of Montpellier for their support, CSHL PacBio sequencing platform. We thank A.R. Krainer, M.L. Hastings, S.

Dokudovskaya, O. Mauger, E. Batsché, O. Espeli, and P. Carroll for the critical reading of the manuscript. We thank L. Telley for his initial contribution to PACBIO analysis. This study was supported by: - The Deutsche Forschungsgemeinschaft (DFG MA6957/1-1), M.M.H. - The ERANet NEURON/DECODE (18-NEUR-0004), F.A. - The REVIVE funding from ANR "Laboratoire d'Excellence" (2011–2021), C.M and E.A. - The ANR KREM-AIF (ANR-21-CE17-0014-02), E.A.

## Author contributions

A.F. and E.A. managed the study; R.D.R., F.A., and E.A. designed the experiments. R.D.R., A.P. J.C., C.J-T., M.M.H., A.S., and E.A. performed the experiments. M.M.H. and A.S. performed the proteomic experiments and A.S., E.A. the analysis. E.K., F.T., C.G., P.N., and E.A. performed PACBIO and Oxford Nanopore sequencing analysis. R.D.R., A.P., and F.A. analyzed the imaging data. E.A. and F.A. wrote the manuscript with contribution from all authors.

## Competing interests

The authors declare no competing interests.
