## [Peer review file · Nature Communications]

REVIEWER COMMENTS

Reviewer #1 (Remarks to the Author):

The authors combine targeted gene approach and long read sequencing to uncover the GPHN isoforms and their differential functions when interaction with inhibitory Glycine and GABA-A receptors. The study shows the relevance of isoform level studies, and how the combinatorial expression pool of the different isoforms of GPHN affects the inhibitory synapses in the cerebellum during mouse brain development and human tissue. The authors identify

The study shows the relevance in the isoform level studies instead of gene level focus uncovering a new catalog of GPHN isoforms and unannotated EEJ, with different expression patterns along development and in different tissues. Moreover, some of these different isoforms are translated in different isoform protein with different function and location in inhibitory synapses. The results are validated in vivo and show evidence in human samples.

It is also interesting to see that all the internal exons from GPHN/Gphn appear as alternative exons. There is some controversy in the field on how to define a constitutive exon, and with more and more data including now the long read technologies is getting more clear that alternative exons are even more abundant than it was thought. Which makes even more difficult to find a truly always constitutive exon.

Comments:

The authors use Gphn targeted approach with PAC-BIO technology to identify all the isoforms on the gene. To validate and compare their sequencing results they authors use a database of published sequencing datasets and find that their approach is able to identify unannotated EEJ. There is currently a quite extensive curated database for EEJ from thousands of datasets, intropolis <https://github.com/nellore/intropolis> . Mainly in human but also has the mouse version of it in snaptron. It would be interesting to compare to this database and recalculate the number of known and novel EEJ recovered with the Gphn targeted approach.

There is an updated database of alternative splicing events that uses public datasets, Vast-db https://vastdb.crg.eu/wiki/Main_Page. Authors should compare non annotated EEJ and alternative exon to the database entries. It would be helpful for to provide the new isoforms and EEJ to public repository and/or provide the annotations the be able to expand the catalog of isoforms in future studies.

The authors mention that Gphn is the first gene with more than 10 exons for which all internal exons can be alternatively spliced. Looking to the isoforms recovered in Fig. S1B seems that for some of the exons are skipped only in 1 out of the 277 isoforms. If the authors look for the expression of those specific isoforms like the case of E2 which is included in one isoform. Does this isoform have enough coverage on E2 and significant number of EEJ?

On the other hand, some exons are included in almost all isoforms the case of E36 for instance. How is the expression of those specific isoforms? Do they have enough coverage. I am guessing that isoforms excluding E36 are very low expressed or/and very specific to a celltype or/and tissue.

Would the authors consider all these very lowly present isoforms as real, leading to a protein or as nuclear noise, or part of the technical limitations of the long-reads technologies, which can lead to noise? Many studies combine bulk sequencing together with long-read to avoid false positives.

All the processing pipeline of the isoform detection is mentioned in the methods, but it should include more details and include the code used. The authors write is a custom pipeline, for that reason I find it more important that this is included in more detail and with the code used. It should include threshold used to consider a significant EEJ and expression levels thresholds to consider a lowly or highly expressed isoform.

In Page 4 authors mention Fig.2D while they are referring to Fig.1D.

In Fig1.F the heatmap shows highly and lowly expressed exons in different celltypes. What is the threshold for defining a high or a low expressed exon? A heatmap showing the actual expression (as a log) of the exon in each celltype will help to show the real expression differences between celltypes.

As I have mentioned before, the authors identify 277 different isoforms. Could be that not all the isoforms are truly expressed in any celltype, and isoform level expression heatmap per celltype and developmental stage will help to clarify if all the detected isoforms are true, at least for these celltypes and developmental stages.

By using a new reference built from the PAC-BIO detected isoforms it would be interesting to look further of the differential isoform usage and differentially spliced events between celltypes using methods like tappAS, <https://app.tappas.org>. Instead of only looking to whole differential expression of general coverage differences between exons.

Since the interest of the study is focus in inhibitory synapses, would be of interest to look differences in different types of neurons at least inhibitory and excitatory. There are available datasets on single cell which classify different neuronal types it would be of interest to see changes in expression of the different isoforms and alternative exons in different types of neurons.

Do the authors have any hint on how these isoforms are conserved in human? With the huge catalog of Gphn isoforms and the functional relevance on synapses connections one could also expect to find a highly alternative spliced and maybe conserved GPHN transcriptome in human. The human GPHN gene already shows many alternative exons as comparison to the mouse Gphn when looking to the datasets in UCSC. In the final part of the manuscript the authors show the detected isoforms in human tissues, which shows many alternative isoforms but less than in mouse. Are any of these isoforms conserved between mouse and human?

In Fig3 C, I am missing the quantification values on the pie-charts. Pie charts are usually confusing to interpret due to the non scaled sizes. Authors should add the proportion for each epitope in the figure. Which will also help to see which ones are the most prevalent combinations, which I cannot clearly see from the pie-charts.

In page 9, cannot find the referred fig.8F

Is it possible to look for known risk SNPs in their PAC-BIO data of each specific, or the validated isoforms? Long reads should give SNP resolution. This would provide more relevance on human disorders and their link to different isoforms.

Authors should include a data availability section with access to the raw sequencing data and processed data used in the study. In the reporting list the ENA accession are included but these should be added to main manuscript. The authors should make a reviewers token to be able to browse the raw and processed data from the study. It would be of huge interest to have the newly annotated reference isoforms and EEJ of GPHN in mouse and human for future studies and increase knowledge.

Authors mentions several RBPs in their conclusions and how they can be regulators of GPHN. There is available data on splicing factors mice KOs and the effect on specific alternative splicing. Considering the isoforms that were validated and follow up in the manuscript, the authors could look for binding of the most typical factors as for instance NOVA,PTBP1/2 around the identified novel EEJ and 3' splice sites and also if there is any effect after KO from published datasets. The results from the manuscript open many

venues on trying to identify the regulators of these specific isoforms and how this affect the final isoform combination in inhibitory synapses diversity and specialization.

Reviewer #2 (Remarks to the Author):

This is an excellent paper of very high quality that provides a comprehensive understanding of Gephyrin molecular and synaptic diversity. Of particular merit is the general systematic nature of the work, the corroboration of mRNA and protein data, the studies of subcellular distribution, the functional consequences of misexpression and the addition of the human studies. This paper makes a significant contribution to our understanding of synapses diversity and the molecular mechanisms underpinning it.

One minor point: the developmental study in figure 1D points to changes in synapse composition and diversity during development and the authors may wish to cite that this has been reported in excitatory synapses using scaffold protein markers using single-synapse resolution data (Cizeron et al, Science 2020 DOI: 10.1126/science.aba3163).

Reviewer #3 (Remarks to the Author):

The enclosed paper does a remarkably thorough analysis of gephrin isoforms across the CNS and associated non-neural tissue. Their results, which is among the first to so fully document the splice forms of a single protein revealed diversity far beyond that was anticipated with an additional undocumented 10 exons being identified. In addition they found large numbers of unexpected forms of gephrin, over half based on Mass Spec appear to actually be produced. Next they study both the cellular and intracellular distributions of these forms and find strong correlations between the distribution both across cell types and within specific cellular compartments (dendrite vs axon). As gephrin exists as a multimeric protein comprised of hetromers of different isoforms they use antibodies against specific epitopes to examine the co-distribution of these four epitopes and find they have different combinations with specific stoichiometry combinations of isomers. While these combinations are not random with regards to compartments or cell types neither are they absolute. Similarly they find the

surprising presence of isoforms thought to be associated with neuropsychiatric conditions are found normally within wild type neurons. On one hand this is a wonderful *tout de force* analysis but on the other the reader is left short of knowing how isoforms relate to function with suggestions but no smoking gun to indicate that these alternatively spliced forms have an impact on actual postsynaptic function of inhibitory synapses. What seemed obviously missing was an attempt to relate isoform prevalence to postsynaptic inhibitory function. While reading this paper I continually puzzled with how such a connection could be made and the best I could come up with was to use GOF of some of the novel isoforms (particularly those that might be associated with neurological dysfunction) to see if they in this context could disrupt inhibitory signaling. Given that they likely do normally in affected patients, this is in part a circular argument, although still GOF disruption would extend causation beyond correlation. All that said the authors have done a truly insane amount of work here and their findings will be of interest to the general public as an exemplar of the need for such analysis to truly understand the impact of isoforms on protein function in general and gephrin in particular. On balance I think this paper stands on its own and while I would encourage them to include some discussion of the need to causal experiments with GOF in the future, I do feel this is too high a bar to demand of them after their already admittedly Herculean efforts.

REVIEWER COMMENTS

Reviewer #1 (Remarks to the Author):

The authors combine targeted gene approach and long read sequencing to uncover the GPHN isoforms and their differential functions when interaction with inhibitory Glycine and GABA-A receptors. The study shows the relevance of isoform level studies, and how the combinatorial expression pool of the different isoforms of GPHN affects the inhibitory synapses in the cerebellum during mouse brain development and human tissue. The authors identify

The study shows the relevance in the isoform level studies instead of gene level focus uncovering a new catalog of GPHN isoforms and unannotated EEJ, with different expression patterns along development and in different tissues. Moreover, some of these different isoforms are translated in different isoform protein with different function and location in inhibitory synapses. The results are validated in vivo and show evidence in human samples.

It is also interesting to see that all the internal exons from GPHN/Gphn appear as alternative exons. There is some controversy in the field on how to define a constitutive exon, and with more and more data including now the long read technologies is getting more clear that alternative exons are even more abundant than it was thought. Which makes even more difficult to find a truly always constitutive exon.

Comments:

The authors use Gphn targeted approach with PAC-BIO technology to identify all the isoforms on the gene. To validate and compare their sequencing results they authors use a database of published sequencing datasets and find that their approach is able to identify unannotated EEJ. There is currently a quite extensive curated database for EEJ from thousands of datasets, intropolis <https://github.com/nellore/intropolis> . Mainly in human but also has the mouse version of it in snaptron. It would be interesting to compare to this database and recalculate the number of known and novel EEJ recovered with the Gphn targeted approach.

Point 1

We would like to thank the reviewer for her/his suggestion. We have now used the mouse version of intropolis to validate new EEJs and the analysis was added to the Fig. 1C and Table S2 of the manuscript. Using intropolis database, we confirmed 11 new EEJs compared to Ensembl database EEJs, however the intropolis database provides less coverage than the dataset of short reads ($2,872.10^9$ short sequences) that we used by compiling public RNAseq projects (Table S2).

The manuscript was modified to incorporate analysis of Gphn expression by using Snaptron. Overall, we observe that our Gphn targeted analysis using PAC-BIO technology far exceeds previous analysis, and therefore promotes a better understanding of the Gphn expression diversity. Not only do we provide a broader description of the multiple EEJs, but we also identify the different transcripts that carry these specific EEJs. Multiple data sets obtained by short-read sequencing technologies are not able to provide such information.

There is an updated database of alternative splicing events that uses public datasets, Vast-db https://vastdb.crg.eu/wiki/Main_Page. Authors should compare non annotated EEJ and alternative exon to the database entries. It would be helpful for to provide the new isoforms and EEJ to public repository and/or provide the annotations the be able to expand the catalog of isoforms in future studies.

Point 2

Thank you for this suggestion, we have used VastDB and incorporated in Fig.S1C the alternative exons annotated in Vasdb for Gphn expression in mouse. Furthermore, Fig.S1C also shows the number of transcripts carrying each alternative exon referenced in VastDB.

Concerning the submission of mouse Gphn transcripts to Vastdb, we have contacted the administrator of this database and it is currently not possible to submit full length transcripts because they are specialized in event-level information.

We agree with the reviewer's point regarding the importance of providing this information to the public. All new Gphn transcripts are referenced in Table S1 and S6 for mouse and human respectively, in addition we now provide BAM and BED files in supplemental information.

The authors mention that Gphn is the first gene with more than 10 exons for which all internal exons can be alternatively spliced. Looking to the isoforms recovered in Fig. S1B seems that for some of the exons are skipped only in 1 out of the 277 isoforms. If the authors look for the expression of those specific isoforms like the case of E2 which is included included in one isoform. Does this isoform have enough coverage on E2 and significant number of EEJ?

Point 3

Thank you for pointing this out.

We performed several tests to confirm that transcripts covered with one read were biologically significant. We performed a manual clearance step of sequences by visual inspection of long read clusters that led to removing one third of CCS. In addition, we show in Fig.S1E, S1F, S1I and S2A that multiple transcripts covered with a single read (Gphn-244, -192, -203) were efficiently validated by RT-PCR. One can note that Gphn-244 expression is restricted to specific tissues (see Fig. 1E), demonstrating that even weakly expressed transcripts are subjected to specific regulation between different mouse tissues. Furthermore, our mass spectrometry analysis validated GPHN protein isoforms translated from transcripts covered with a single sequencing read, for instance GPHN-226, -235, -269 (Table S5). Overall we can conclude that transcripts covered with one single read are sustained by biological evidence and validates our claim about GPHN splicing regulation.

On the other hand, some exons are included in almost all isoforms the case of E36 for instance. How is the expression of those specific isoforms? Do they have enough coverage. I am guessing that isoforms excluding E36 are very low expressed or/and very specific to a celltype or/and tissue.

Point 4

We found seven transcripts without E36 and they are weakly represented in our analysis (Gphn-152, Gphn-193, Gphn-198, Gphn-230, Gphn-232, Gphn-240, Gphn-246). We show below the architecture of these transcripts and their regulations in different developmental stages and brain tissues. One can note that their expression is restricted to specific developmental points, suggesting fine tuning of alternative splicing during development. We

would also like to point out that experiments were performed with bulk samples, so it is possible that the expression of these transcripts is much higher in specific subpopulations of neuronal cells as pointed out by the reviewer.

Would the authors consider all these very lowly present isoforms as real, leading to a protein or as nuclear noise, or part of the technical limitations of the long-reads technologies, which can lead to noise? Many studies combine bulk sequencing together with long-read to avoid false positives.

Point 5

This point is reminiscent of point 3, and all our data sustain that diversity of Gphn expression does not result from transcriptional/splicing noise. Following the suggestion of the reviewer in point 1, 2 and 3, we have already compared our data to a large database of bulk sequencing dataset provided by many laboratories around the world. However, such a comparative approach remains poorly efficient because these data were generated by global assessment of gene expression in contrast to our targeted approach. The depth obtained from global gene expression analysis fails to detect under-represented EEJs. In addition, short read sequencing is unable to provide the exonic architecture composing full length transcripts, since GPHN exons and/or EEJs are used by multiple isoforms, as we show in Fig.1C and 8C for mouse and human respectively.

All the processing pipeline of the isoform detection is mentioned in the methods, but it should include more details and include the code used. The authors write is a custom pipeline, for that reason I find it more important that this is included in more detail and with the code used. It should include threshold used to consider a significant EEJ and expression levels thresholds to consider a lowly or highly expressed isoform.

Point 6

We completely agree with the point raised by the reviewer and therefore we have updated the methods section with more details and especially specified some missing parameters/threshold (see section "PacBio, ONT and short reads analysis" of the revised manuscript). It should be noted that our analysis includes several steps that were handled

manually (Fig.S1A), such as the annotation of exons, and we were not able to automate these steps.

Regarding the question about "threshold used to consider a significant EEJ", we add additional information in the Material and Methods section and answer in Point 3.

In Page 4 authors mention Fig.2D while they are referring to Fig.1D.

Point 7

Thanks to the reviewer, this was corrected in the revised version of the manuscript.

In Fig1.F the heatmap shows highly and lowly expressed exons in different celltypes. What is the threshold for defining a high or a low expressed exon? A heatmap showing the actual expression (as a log) of the exon in each celltype will help to show the real expression differences between celltypes.

Point 8

We would like to thank the reviewer for this suggestion. We added a new heatmap version of the Fig1.F in which each exon relative expression in each cell-type is displayed.

As I have mentioned before, the authors identify 277 different isoforms. Could be that not all the isoforms are truly expressed in any celltype, and isoform level expression heatmap per celltype and developmental stage will help to clarify if all the detected isoforms are true, at least for these celltypes and developmental stages.

Point 9

Some of the information requested by the reviewer was already present in the manuscript. We now update the Fig.1D by providing a heatmap of the expression of all isoforms during 4 developmental time points in the cerebellum and the cortex.

Regarding the expression of individual isoforms in different neuronal cell types, we cannot provide this information because the public data used in Fig.1F are from short-read sequencing that ignores exon architecture of individual full-length isoforms. Nevertheless Fig.1F shows that the expression level of individual exons across cell-type is different, which suggests a cell-type dependent regulation of their alternative splicing. Furthermore, this is in agreement with our finding that all internal exons of Gphn are regulated by alternative splicing.

A methodological approach that combines long-read sequencing and single cell analysis will require significant development, and we believe that it is out of the scope of our manuscript.

By using a new reference built from the PAC-BIO detected isoforms it would be interesting to look further of the differential isoform usage and differentially spliced events between celltypes using methods like tappAS, <https://app.tappas.org>. Instead of only looking to whole differential expression of general coverage differences between exons.

Following the reviewer's suggestion, we invested a lot of time and energy to get tappAS working with our data. We provide below to the attention of the reviewer the few results that we succeed to generate with tappAS:

We believe that these data do not make critical changes to the message and analysis that we have already included in the manuscript.

Since the interest of the study is focus in inhibitory synapses, would be of interest to look differences in different types of neurons at least inhibitory and excitatory. There are available datasets on single cell which classify different neuronal types it would be of interest to see changes in expression of the different isoforms and alternative exons in different types of neurons.

Point 11

This point is similar to point 9. Most studies use methods such as "drop seq" or "10x genomics" coupled to short read sequencing, which enable high throughput with a relatively shallow sequencing depth, thus global RNA expression in individual cells is only acquired for most expressed genes making a deep analysis of Gphn splicing regulation redibitory. Supporting our conclusion, the use of a large sequencing dataset corresponding to global gene expression analysis is not sufficient to reveal a lot of EEJ Gphn as indicated in point 1, either with the public databases selected by us or using intropolis.

Do the authors have any hint on how these isoforms are conserved in human? With the huge catalog of Gphn isoforms and the functional relevance on synapses connections one could also expect to find a highly alternative spliced and maybe conserved GPHN transcriptome in human. The human GPHN gene already shows many alternative exons as comparison to the mouse Gphn when looking to the datasets in UCSC. In the final part of the manuscript the authors show the detected isoforms in human tissues, which shows many alternative isoforms but less than in mouse. Are any of these isoforms conserved between mouse and human?

Point 12

We agree with the reviewer that making a comparison between human and mouse is informative, this was already provided in Fig.S5C and commented in the text. To make this point more clear, we added the following text in the manuscript: "Our analysis showed that 28 exons with more than 95% homology are shared between mouse and Human. It is interesting to note that most of the nearly identical exons are the ones that are overrepresented in the transcriptome (Fig.S5B). This observation highly suggests that a significant proportion of mouse and human GPHN isoforms shared the same core protein domains. However, 12 and 14 dissimilar exons are also present in mouse and human respectively, indicating that significant differences also exist."

Regarding the number of alternative transcripts in mouse and human, the reviewer might have missed in the Fig.7C that the GPHN alternative transcripts are far more numerous in any human tissues than in the mouse brain.

In Fig3 C, I am missing the quantification values on the pie-charts. Pie charts are usually confusing to interpret due to the non scaled sizes. Authors should add the proportion for each epitope in the figure. Which will also help to see which ones are the most prevalent combinations, which I cannot clearly see from the pie-charts.

Point 13

Thank you for this suggestion. We have added the proportion of each epitope in the revised Fig.3C.

In page 9, cannot find the referred fig.8F

Point 14

We apologize for this error and this was corrected by replacing Fig.8F with Fig.8A in the revised manuscript.

Is it possible to look for known risk SNPs in their PAC-BIO data of each specific, or the validated isoforms? Long reads should give SNP resolution. This would provide more relevance on human disorders and their link to different isoforms.

Point 15

Human RNA samples used to sequence GPHN transcriptomes were prepared by pooling together 1 to 64 samples and were performed with Oxford Nanopore technologies; we were not able to identify SNPs properly with such a variety of samples and with the error rate of the sequencing technology at the time we engaged these experiments. Moreover, none of the individuals included in these cohorts were diagnosed for neuronal disorders, so in our condition it is not possible to identify SNP that will be relevant in human disorders.

Authors should include a data availability section with access to the raw sequencing data and processed data used in the study. In the reporting list the ENA accession are included but these should be added to main manuscript. The authors should make a reviewers token to be able to browse the raw and processed data from the study. It would be of huge interest to have the newly annotated reference isoforms and EEJ of GPHN in mouse and human for future studies and increase knowledge.

Point 16

A chapter "Raw sequencing data, BAM and BED files" are available in the revised manuscript at the section Materials and Methods.

Authors mentions several RBPs in their conclusions and how they can be regulators of GPHN. There is available data on splicing factors mice KOs and the effect on specific alternative splicing. Considering the isoforms that were validated and follow up in the manuscript, the authors could look for binding of the most typical factors as for instance NOVA,PTBP1/2 around the identified novel EEJ and 3'splice sites and also if there is any effect after KO from published datasets. The results from the manuscript open many venues on trying to identify the regulators of these specific isoforms and how this affect the final isoform combination in inhibitory synapses diversity and specialization.

Point 17

In published datasets, the authors have essentially examined the seven gephyrin exons reported to be alternatively spliced. For example, in NOVA knockout mice, they found that only exon 9 was significantly regulated. Based on our study, the exon 9 (exon 17 in the new annotation) is included in 230 transcripts (check the Fig.S1C), it is therefore not possible to identify isoforms regulation with these data sets. To address this question, one will need to perform a new study by combining Gphn-targeted long read sequencing in different mutant mouse models such as Nova or PTBP1/2 mouse models. Nevertheless, we agree with the reviewer that it is an important issue that will need to be addressed in the future, as we noted in our conclusion.

Reviewer #2 (Remarks to the Author):

This is an excellent paper of very high quality that provides a comprehensive understanding of Gephyrin molecular and synaptic diversity. Of particular merit is the general systematic nature of the work, the corroboration of mRNA and protein data, the studies of subcellular distribution, the functional consequences of misexpression and the addition of the human studies. This paper makes a significant contribution to our understanding of synapses diversity and the molecular mechanisms underpinning it.

One minor point: the developmental study in figure 1D points to changes in synapse composition and diversity during development and the authors may wish to cite that this has been reported in excitatory synapses using scaffold protein markers using single-synapse resolution data (Cizeron et al, Science 2020 DOI: 10.1126/science.aba3163).

Point 18

We would like to thank the reviewer for the glowing superlatives he wrote about our manuscript and our study. We agree that this work is important for the field of inhibitory synapse diversity and we thank him/her for the kind comments. We really appreciate it. The reference is now added in the text page 4 - line 28

Reviewer #3 (Remarks to the Author):

The enclosed paper does a remarkably thorough analysis of gephrin isoforms across the CNS and associated non-neural tissue. Their results, which is among the first to so fully document the splice forms of a single protein revealed diversity far beyond that was anticipated with an additional undocumented 10 exons being identified. In addition they found large numbers of unexpected forms of gephrin, over half based on Mass Spec appear

to actually be produced. Next they study both the cellular and intracellular distributions of these forms and find strong correlations between the distribution both across cell types and within specific cellular compartments (dendrite vs axon). As gephrin exists as a multimeric protein comprised of heteromers of different isoforms they use antibodies against specific epitopes to examine the co-distribution of these four epitopes and find they have different combinations with specific stoichiometry combinations of isomers. While these combinations are not random with regards to compartments or cell types neither are they absolute. Similarly they find the surprising presence of isoforms thought to be associated with neuropsychiatric conditions are found normally within wild type neurons. On one hand this is a wonderful *tout de force* analysis but on the other the reader is left short of knowing how isoforms relate to function with suggestions but no smoking gun to indicate that these alternatively splice forms have an impact on actual postsynaptic function of inhibitory synapses. What seemed obviously missing was an attempt to relate isoform prevalence to postsynaptic inhibitory function. While reading this paper I continually puzzled with how such a connection could be made and the best I could come up with was to use GOF of some of the novel isoforms (particularly those that might be associated with neurological dysfunction) to see if they in this context could disrupt inhibitory signaling. Given that they likely do normally in affected patients, this is in part a circular argument, although still GOF disruption would extend causation beyond correlation. All that said the authors have done a truly insane amount of work here and their findings will be of interest to the general public as an exemplar of the need for such analysis to truly understand the impact of isoforms on protein function in general and gephrin in particular. On balance I think this paper stands on its own and while I would encourage them to include some discussion of the need for causal experiments with GOF in the future, I do feel this is too high a bar to demand of them after their already admittedly Herculean efforts.

Point 19

First, we would like to thank the reviewer for finding our work of interest for the general public and we appreciate his/her kind words about our work. We completely agree that in the future, we will need to assess in more detail the role of GPHN variants in inhibitory synapse signaling. Maybe the reviewer missed this part in our manuscript but we performed several isoforms GOF in cerebellum and observed important effects on inhibitory synapse formation/stabilization (Fig.5) Following his/her recommendation, we added a new paragraph in the discussion to highlight the need of causal experiments with GOF in the future.

REVIEWERS' COMMENTS

Reviewer #1 (Remarks to the Author):

Thanks to the authors for considering my comments. I see the changes and additions have increased the value of the study, which already showed good quality and insights. I find it a great paper that shows the effort and good work of the authors. I am totally satisfied with the revisions and I would recommend it for acceptance.